# Inhibition of BK channels by GABAb receptors enhances intrinsic excitability of layer 2/3 vasoactive intestinal polypeptide-expressing interneurons in mouse neocortex

Karolina Bogaj (ID) and Joanna Urban-Ciecko (ID)

*Laboratory of Electrophysiology, Nencki Institute of Experimental Biology, Polish Academy of Sciences, Warsaw, Poland*

Handling Editors: Katalin Toth & Jean-Claude Béïque

The peer review history is available in the Supporting Information section of this article (https://doi.org/10.1113/JP286439#support-information-section).

**Abstract figure legend** Using *in vitro* whole-cell patch-clamp recordings in layer 2/3 of the mouse somatosensory cortex, three main electrophysiological clusters of vasoactive intestinal polypeptide-expressing interneurons (VIP-INs) were found. Baclofen, an agonist of GABAbRs, had different effects on VIP-IN intrinsic excitability, depending on [Ca$^{2+}$] in extracellular solution. In high [Ca$^{2+}$] (2.5 mM), postsynaptic GABAbRs indirectly inhibit BK channels and activate GIRK channels, whereas presynaptic GABAbRs suppress GABA release and thus reduce postsynaptic GABAaR inhibition. In this condition, activation of GABAbRs leads to increased excitability of type 1 VIP-INs. However, in more physiological (low, 1 mM) [Ca$^{2+}$], BK channels do not regulate the intrinsic excitability of VIP-INs and thus postsynaptic GABAbRs canonically decrease the intrinsic excitability of type 1 through GIRK channels.

**Abstract** GABAb receptors (GABAbRs) affect many signalling pathways, and hence the net effect of the activity of these receptors depends upon the specific ion channels that they are linked to, leading to different effects on specific neuronal populations. Typically, GABAbRs suppress neuronal activity in the cerebral cortex. Previously, we found that neocortical parvalbumin-expressing cells are strongly inhibited through GABAbRs, whereas somatostatin interneurons are immune to this modulation. Here, we employed *in vitro* whole-cell patch-clamp recordings to study whether GABAbRs modulate the activity of vasoactive intestinal polypeptide-expressing interneurons (VIP-INs) in layer (L) 2/3 of the mouse primary somatosensory cortex. Utilizing machine learning algorithms (hierarchical clustering and principal component analysis), we revealed that one VIP-IN cluster (about 68% of all VIP-INs) was sensitive to GABAbR activation. Paradoxically, when recordings were performed in standard conditions with high extracellular $Ca^{2+}$ level, GABAbRs indirectly inhibited the activity of large conductance voltage- and calcium-activated potassium (BK) channels and reduced GABAaR-mediated inhibition, leading to an increase in intrinsic excitability of these interneurons. However, a classical inhibitory effect of GABAbRs on L2/3 VIP-INs was observed in modified artificial cerebrospinal fluid with physiological (low) $Ca^{2+}$ concentration. Our results are essential for a deeper understanding of mechanisms underlying the modulation of cortical networks.

(Received 16 February 2024; accepted after revision 16 January 2025; first published online 4 February 2025)

**Corresponding author** J. Urban-Ciecko: Laboratory of Electrophysiology, Nencki Institute of Experimental Biology, Polish Academy of Sciences, Warsaw 02-093, Poland. Email: j.ciecko@nencki.edu.pl

## Key points

- Layer 2/3 vasoactive intestinal polypeptide-expressing interneurons (VIP-INs) in the mouse somatosensory cortex cluster into three electrophysiological types differentially sensitive to GABAb receptors (GABAbRs).
- The majority of VIP-INs (type 1, about 68% of all VIP-INs) are regulated through pre- and post-synaptic GABAbRs, while a subset of these interneurons (types 2 and 3) is controlled only pre-synaptically.
- The net effect of GABAbR activation on VIP-IN excitability depends on $[Ca^{2+}]$ in artificial cerebrospinal fluid.
- When $[Ca^{2+}]$ is high (2.5 mM), GABAbRs indirectly inhibit BK channels and reduce GABAaR inhibition leading to increased intrinsic excitability of type 1 VIP-INs.
- When $[Ca^{2+}]$ is low (1 mM), which is more physiological, BK channels do not regulate the intrinsic excitability of VIP-INs and thus postsynaptic GABAbRs canonically decrease the intrinsic excitability of type 1 VIP-INs.

## Introduction

The inhibitory neurotransmitter GABA shapes the activity of cortical networks and supports cognitive functions such as learning and memory (Gassmann & Bettler, 2012; Padgett & Slesinger, 2010). GABA is released by a population of highly diverse GABAergic interneurons

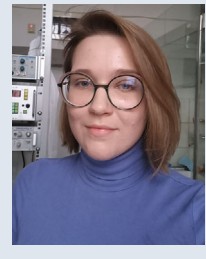

**Karolina Bogaj** received her BSc degree in Animal Bioengineering and MSc degree in Biotechnology. She is currently a Research Assistant pursuing a PhD in Neuroscience at the Nencki Institute of Experimental Biology, Warsaw, Poland. Using *in vitro* electrophysiology, she aims to investigate local neuronal circuitry within the rodent somatosensory cortex.

(Booker et al., 2018, 2020; Degro et al., 2015; Kanigowski et al., 2023; Urban-Ciecko et al., 2010). Among them, vasoactive intestinal polypeptide-expressing interneurons (VIP-INs) account for approximately 10–15% of GABAergic interneurons in the neocortex (Prönneke et al., 2015; Xu et al., 2010). In the mouse somatosensory cortex, the vast majority of VIP-INs can be found in layer 2/3 (~60%) (Almási et al., 2019; Lee et al., 2010; Prönneke et al., 2015; Xu et al., 2010). VIP-INs belong to a subgroup of inhibitory interneurons expressing the serotonin 5-hydroxytryptamine 3a receptor (5HT3aR), non-overlapping with interneurons expressing somatostatin (SST) or parvalbumin (PV) (Almási et al., 2019; Lee et al., 2010; Prönneke et al., 2015; Xu et al., 2010).

VIP-INs are activated during high arousal states, suggesting that these interneurons contribute to state-dependent modulation of local circuit activity (Dipoppa et al., 2018; Fu et al., 2014; Pakan et al., 2016). It has been established that VIP-INs play an important role in the neocortex, creating a disinhibitory circuit by targeting other inhibitory interneurons (predominantly SST-INs) and thus releasing pyramidal cells from inhibition (Jiang et al., 2015; Lee et al., 2013; Pfeffer et al., 2013). This disinhibitory loop is essential in sensory information processing (Ferguson et al., 2023) and during the learning process (Keller et al., 2020; Letzkus et al., 2015; Myers-Joseph et al., 2023). VIP-IN dysfunction has serious consequences, since in a rodent model, an experimental disruption of VIP-IN function phenocopies behavioural abnormalities identified in autism spectrum disorder (Goff et al., 2023) and intellectual disabilities (Goff & Goldberg, 2021; Mossner et al., 2020).

The activity of L2/3 VIP-INs is modulated by long-range inputs (Lee et al., 2013; Wall et al., 2016), as well as by local GABAergic interneurons (Pfeffer et al., 2013). Previously, we have shown that L2/3 VIP-INs are inhibited by tonic and phasic inhibition mediated by ionotropic GABA receptors (GABAaRs) (Bogaj et al., 2023). However, the modulation of neocortical VIP-IN activity through metabotropic GABA receptors (GABAbRs) remains largely unknown.

GABAbRs are G-protein-coupled metabotropic receptors and may be expressed pre- and postsynaptically on both excitatory and inhibitory neurons (Booker et al., 2018, 2020; Degro et al., 2015; Kanigowski et al., 2023; Urban-Ciecko et al., 2010). These receptors can affect many signalling pathways, and hence the net effect of GABAbR activity will depend upon the specific ion channels that they are linked to, causing different effects on specific neuronal populations. Typically, postsynaptic GABAbRs suppress neuronal activity through the activation of G-protein-coupled potassium (GIRK) channels (Gähwiler & Brown, 1985; Mintz & Bean, 1993), whereas presynaptic GABAbRs inhibit $Ca^{2+}$ channels and thus reduce neurotransmitter release (Mintz & Bean, 1993). However, also excitatory effects of GABAbRs have been recognized (Garaycochea & Slaughter, 2016; Ramakrishna & Sadeghi, 2020), especially when GABAb autoreceptors suppress GABA release.

Growing evidence indicates different functions of GABAbRs on inputs and outputs of specific neuronal populations (Booker et al., 2017, 2018, 2020; Kanigowski et al., 2023; Urban-Ciecko et al., 2015). For example, the intrinsic excitability of L2/3 PV-INs is reduced by postsynaptic GABAbRs whereas L2/3 SST-INs are completely immune to the modulation by these receptors (Kanigowski et al., 2023).

In the rat hippocampus, GABAbRs colocalize with cholecystokinin (CCK)-expressing interneurons (Booker et al., 2016). A fraction of CCK cells co-expresses VIP (VIP[+]/CCK[+]-INs), contributing to a rare subgroup of VIP-INs (Georgiou et al., 2022; Hioki et al., 2018; Kubota et al., 2011; Rhomberg et al., 2018; Tremblay et al., 2016) and suggesting that at least some of the VIP-INs express GABAbRs.

In this study, using *in vitro* whole-cell patch-clamp recordings and pharmacological tools, we asked whether GABABBRs modulate the activity of L2/3 VIP-INs in the mouse somatosensory (barrel) cortex. Literature data show that VIP-INs are extremely heterogeneous in their morphological, electrophysiological and molecular features (Jiang et al., 2023; Prönneke et al., 2015, 2019). For this reason, we first clustered these interneurons using unsupervised and supervised machine-learning methods such as hierarchical clustering, random forest classifier and principal component analysis (PCA) and then asked whether these neuronal clusters are sensitive to GABAbR modulation. Our study for the first time shows functional evidence of the presence of GABAbR on VIP-INs in layer 2/3 of the rodent somatosensory cortex. Nevertheless, sensitivity to modulation through GABAbRs depends on the cluster type, since only one out of three VIP-IN clusters was sensitive to the GABAbR agonist. We uncovered that GABAbRs regulate VIP-IN activity through several mechanisms. In the condition of a high extracellular $Ca^{2+}$ level (2.5 mM), GABAbRs increase the intrinsic excitability of a subset of L2/3 VIP-INs, indirectly through an inhibition of large conductance voltage- and calcium-activated potassium (BK) channels. Additionally, our data suggest that the activation of GABAbRs increases L2/3 VIP-IN intrinsic excitability by suppressing GABA release and thus reducing GABAaR-mediated inhibition. However, in the condition of a physiological (low) $Ca^{2+}$ level (1 mM), a canonical inhibitory effect of GABAbRs on VIP-IN activity can be revealed.

## Methods

### Animals

Experiments were conducted in all respects according to the European Community Council Directive (86/609/EEC) regarding transgenic mice usage in research and in accordance with the Act on the Protection of Animals Used for Scientific or Educational Purposes in Poland (Act of 15 January 2015, changed 17 November 2021; directive 2010/63/EU). All the procedures were approved by the Institutional Animal Care and Use at the Nencki Institute.

Animal strains were acquired from The Jackson Laboratory (Bar Harbor, ME, USA), stock no. 010908 (VIP-CRE; C57Bl6J background) and no. 007908 (Ai14; C57Bl6J and B6 mixed background). VIP-CRE homozygous females were bred with Ai14 (Tdt-floxed) homozygous males, generating heterozygous VIP-CRE::Ai14 offspring.

At P21 mice were weaned and housed in cages with *ad libitum* food and water and a 12 h day–night cycle. Animals of both sexes were used for the experiments.

### Acute brain slices preparation

Mice aged P21–P28 were deeply anaesthetized with isoflurane and decapitated. Brains were extracted and acute brain slices were prepared in an oblique coronal plane (45–55°) across the barrel field (Urban-Ciecko et al., 2015, 2018), and sectioned to 350 µm thickness with a vibrating-blade microtome. Slices were cut in chilled (approx. 4°C) regular artificial cerebrospinal fluid (rACSF), then recovered and maintained in warm (approx. 31°C in a custom-made incubation chamber with temperature-controlled heating plate) regular or modified ACSF (mACSF), depending on the experiment. Regular ACSF was composed of (in mM): 113 NaCl, 2.5 KCl, 1.3 $MgSO_4$, 2.5 $CaCl_2$, 1 $NaH_2PO_4$, 26.2 $NaHCO_3$ and 11 glucose, equilibrated with 95% $O_2$–5% $CO_2$ for pH adjustment. In mACSF, the following concentrations (in mM) were used: 113 NaCl, 3.5 KCl, 0.5 $MgSO_4$, 1 $CaCl_2$, 1 $NaH_2PO_4$, 26.2 $NaHCO_2$, 11 glucose (Urban-Ciecko et al., 2015). Slices were allowed 2–5 h for incubation prior to transportation to the microscope recording chamber, perfused with warm and carbogen-bubbled ACSF. Whole-cell patch-clamp recordings were performed at 31–32°C. ACSF was heated using an inline heater with a temperature control system (Harvard Apparatus, Holliston, MA, USA).

### Electrophysiological recordings *in vitro*

The barrel field of the primary somatosensory cortex was localized under low magnification, whereas a single neuronal soma was identified using a ×40 water-immersion lens. L2/3 VIP-INs were identified using a fluorescent reporter gene expressed in VIP-CRE::Ai14 mice. Recordings were obtained via borosilicate glass electrodes (4–7 MΩ resistance) when filled with an internal solution consisting of (mM): 125 potassium gluconate, 2 KCl, 10 Hepes, 0.5 EGTA, 4 MgATP and 0.3 NaGTP, at pH 7.2, 270 mOsm (Urban-Ciecko et al., 2010, 2015, 2018). Signals were acquired and filtered at 3 kHz with a Multiclamp 700B patch-clamp amplifier, digitized at 20 kHz using a Digidata 1550B interface and collected with pClamp 10 software (Molecular Devices, San Jose, CA, USA).

Access resistance was monitored throughout the entire recording. Recordings with access resistance either exceeding 35 MΩ or changed by >30% were discarded from further analysis. To measure input resistance ($R_{in}$), a test pulse (−10 pA lasting 100 ms) was applied at the beginning of every sweep in the current-clamp mode. Access resistance, membrane capacitance (Cp) and membrane time constant (tau) were measured automatically using a membrane test tool in pClamp (test pulse of −10 mV lasting 30 ms).

Intrinsic excitability was analysed in the current-clamp mode in response to a 500 ms square pulse of an increasing step amplitude of 10 pA. For intrinsic excitability measurements, membrane potential was maintained at −70 mV in different pharmacological conditions. Spontaneous action potential firings (sAPs) were measured in the current-clamp mode at resting membrane potential in mACSF. Spontaneous excitatory postsynaptic currents (sEPSCs) were measured in the voltage-clamp mode at the holding potential of −70 mV in rACSF.

### *In vitro* pharmacology

Drugs used for the study were as follows: GABAbR agonist baclofen, 10 µM (Sigma-Aldrich, St Louis, MO, USA), NMDA receptor antagonist D-2-amino-5-phosphonovaleric acid (APV), 50 µM (Tocris Bioscience, UK), AMPA receptor antagonist 6,7-dinitroquinoxaline-2,3-dione (DNQX), 20 µM (Tocris Bioscience, Bristol, UK), BK channel antagonist paxilline, 10 µM (Tocris Bioscience), GABAaR antagonist picrotoxin, 100 µM (Tocris Bioscience).

Pharmacological agents were bath-applied for no less than 10 min with 3 ml/min perfusion flow before the acquisition of drug effects.

### Data analysis

All the analyses of electrophysiological parameters were carried out using Clampfit software (Molecular Devices). sAPs were detected with a threshold-based search tool,

with the criterion of spikes overshooting a 0 mV threshold. sEPSCs were analysed with a template search tool, in which templates were manually fitted to identify events, with cutoff at 5 pA amplitude. The event frequency of sEPSCs or sAPs was determined as the number of events within a 1 s time period (Hz). Sag amplitude, which is a measure of $I_h$ current, was calculated as a difference between minimum and steady-state values of the voltage in the hyperpolarizing square pulse ($-200$ pA, 500 ms). With the same membrane hyperpolarizing pulse, we checked the presence of a rebound spike. The membrane time constant (tau) was measured from rising and decay phases of a single exponential curve best fitted to the transient current response evoked by the hyperpolarizing voltage step ($-10$ mV, 30 ms). Tau was the time for the current to reach 63% of its steady-state value.

Maximum spike frequency was calculated as the maximum spike number in response to incremental depolarizing current steps within a 500 ms square pulse (Hz), with cutoff set at 5 mV amplitude. Mean frequency was calculated as averaged firing frequency across all steps. Rheobase was determined as the minimum current step in which spikes appeared for the first time. Properties of AP shape were examined at a current step evoking around 50% of the maximum firing, using the threshold search tool in Clampfit. The spike threshold was determined as a value of voltage when the AP rising phase accelerated to 5 mV/ms. AP half-width was calculated as a time difference between the rising and decay phase of the spike at half-maximum amplitude. The amplitude of fast and medium afterhyperpolarization (AHP) was measured as a difference in voltage between the AP threshold and the lowest point of AHP at 4 and 6 ms, respectively. Mean interevent interval (ISI) was defined as the mean time between AP peaks in a given step. Change in amplitude was calculated as the difference between amplitudes of the second and the first spike divided by the amplitude of the first spike. Midpoint and steepness were calculated based on a fitted curve, according to the formula in the section below.

## Curve fitting

For the analysis of intrinsic excitability, a firing frequency-to-current intensity plot (F-I curve) was created for each neuron in the control condition and after drug administration. Subsequently, using the same raw data, a sigmoidal curve was fitted to every F-I curve employing the sigmoidal Hill function with the following equation:

$$y = \frac{a \times x^b}{c^b + x^b}$$

As a result, this curve fitting analysis provided three parameters describing dynamics of intrinsic excitability: (a) maximum of a curve (maximum frequency), (b) steepness of a slope defined as neuronal gain, and (c) midpoint of a curve, a point on the *x*-axis describing half of the maximum frequency. Curve fitting analysis was performed in Python.

## Clustering

Neurons were clustered based on 18 electrophysiological features: rheobase, membrane capacitance, resting membrane potential, input resistance, maximum spike frequency, sag amplitude, presence of rebound spike, burst or adaptive spiking, mean firing frequency, medium and fast AHP, midpoint and steepness of the F-I curve, membrane time constant (tau), spike threshold, spike amplitude change, AP half-width, and mean interevent interval (ISI). Data were hierarchically clustered and visualized as a dendrogram, using Ward's method (Ward, 1963). As a result, neurons were grouped into three clusters (types). For better characterization of types, we applied a supervised machine learning tool – Random Forest Classifier (Breiman, 2001) – which predicts the data categorization after training on the dataset to assess the variable contribution to cluster allocation. Next, to better understand which membrane properties are crucial for each cluster assignment, the dataset was dimensionally reduced using PCA (Wold et al., 1987). Then we plotted a vector factor map showing relations between all electrophysiological properties in the PCA matrix. Vector length and direction represent feature correlation with the principal component. Vectors with opposite directions are negatively correlated to each other, whereas the closer the vectors are to each other, the more correlated are the related properties (Jolliffe & Cadima, 2016). Clustering analysis and visualization were performed using the original Python script.

## Statistical analysis

All data are presented as mean with standard deviation (SD). Data were initially tested for normal distribution with the Shapiro–Wilk test. Statistical significance was defined as $P < 0.05$ using Student's paired *t* test, Wilcoxon's signed-rank test, a two sample *t* test, the Kruskal–Wallis test with Dunn's *post hoc* test or one-way ANOVA test with Holm–Šidák *post hoc* test, as indicated. Statistical analysis and visualization were performed using the Python environment and SigmaPlot (Grafiti LLC, Palo Alto, CA, USA).

## Results

### L2/3 VIP-IN intrinsic excitability increases after the activation of GABAbRs at high extracellular Ca²⁺ level

To study L2/3 VIP-INs, we used *in vitro* whole-cell patch-clamp recordings of fluorescently labelled VIP-INs located in the primary somatosensory (barrel) cortex of transgenic mice. Taking into account the tremendous diversity of L2/3 VIP-INs, we first characterized these neurons on the electrophysiological level employing the unsupervised machine-learning method – hierarchical clustering (Fig. 1) – as proposed by Jiang and colleagues (Jiang et al., 2023). This analysis allowed us to classify L2/3 VIP-INs into specific subpopulations (types). As

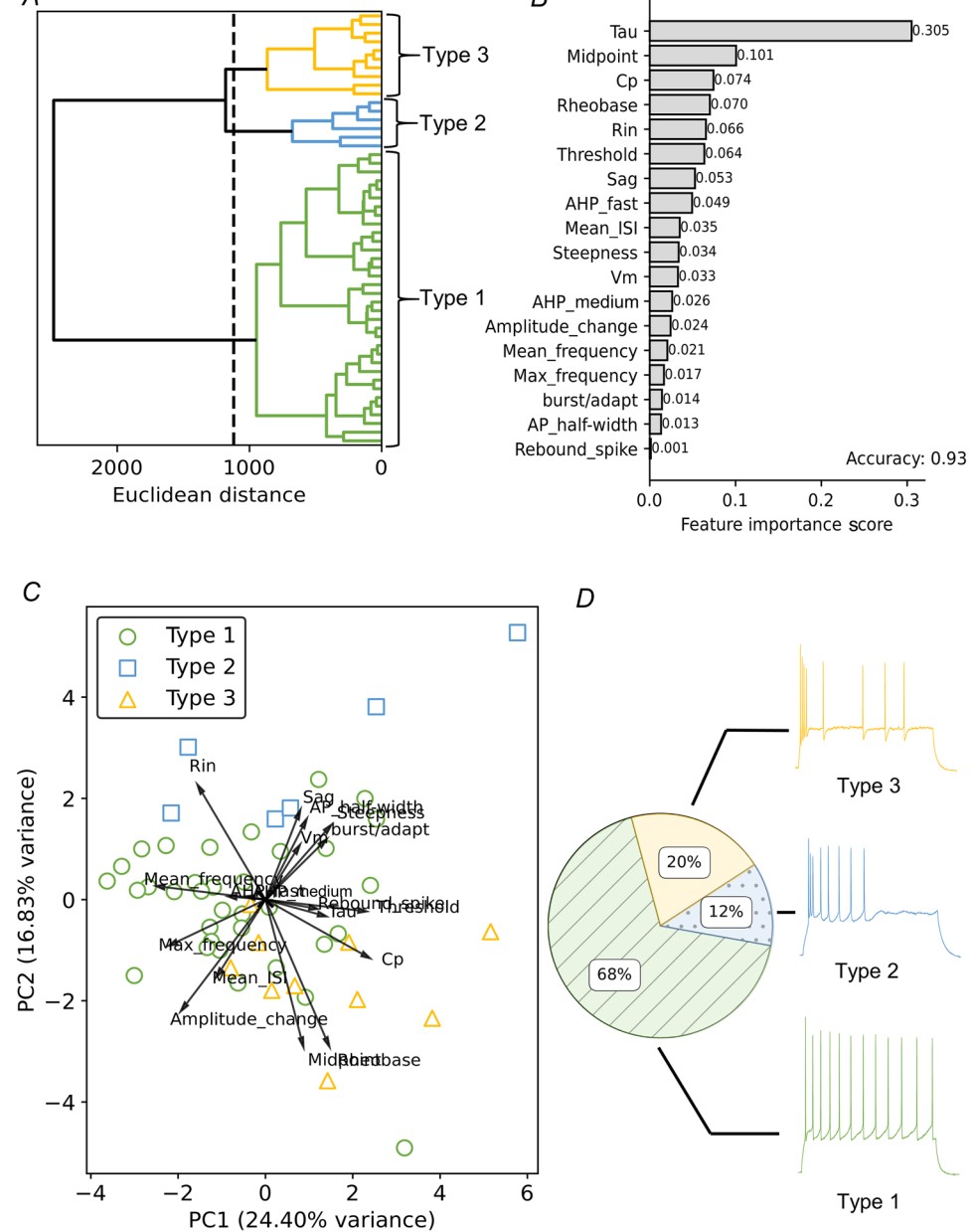

**Figure 1. Electrophysiological classification of L2/3 VIP-INs**
*A*, dendrogram obtained using Ward's method based on electrophysiological properties, an unsupervised machine learning method. Dashed line intersects branches that represent each cluster. *B*, random forest feature importance graph showing influence of each membrane property on cluster assignment. *C*, principal component analysis (PCA) graph with vector factor map indicating features that shape cluster distribution of 3 VIP-IN types according to their electrophysiological properties (*n* = 50 cells in 37 mice). *D*, pie chart of interneuron type distribution (left) and examples of representative firing patterns (right). [Colour figure can be viewed at wileyonlinelibrary.com]

**Table 1. Membrane properties before and after the GABAbR agonist (baclofen) application in rACSF**

| VIP-IN type | Membrane potential (mV) | | | Input resistance (MΩ) | | |
|---|---|---|---|---|---|---|
| | Ctrl | Bac | *P*-value (*n* cells) | Ctrl | Bac | *P*-value (*n* cells) |
| Type 1 | $-62.44 \pm 6.49$ | $-65.35 \pm 5.50$ | <0.001* (34) Wilcoxon test | $428.42 \pm 133.34$ | $403.85 \pm 129.11$ | 0.002* (34) paired *t* test |
| Type 2 | $-60.56 \pm 6.67$ | $-63.59 \pm 4.69$ | 0.156 (6) Wilcoxon test | $652.06 \pm 179.13$ | $558.26 \pm 141.54$ | 0.066 (6) paired *t* test |
| Type 3 | $-63.94 \pm 6.14$ | $-68.21 \pm 4.39$ | 0.002* (10) paired *t* test | $312.02 \pm 90.43$ | $286.12 \pm 98.41$ | 0.026* (10) paired *t* test |

Data are means $\pm$ SD. *Statistical significance.

described previously (Ascoli et al., 2008; Jiang et al., 2023; Prönneke et al., 2015, 2019), we observed a variety of firing patterns of L2/3 VIP-INs (Fig. 1*D*). For cluster distribution, first we utilized Ward's method, obtaining a cluster dendrogram (Fig. 1*A*) based on 18 fundamental intrinsic properties of VIP-INs (Fig. 1*C*). Dendrogram tree analysis showed three main clusters of L2/3 VIP-INs similarly to what has been found by Jiang et al. (2023). Random Forest Classifier, a supervised learning algorithm, was trained to estimate pivotal features for dendrogram cluster assignment (Fig. 1*B*). With accuracy reaching 93%, the model evaluated membrane time constant (tau) as the most important factor for cluster distribution (0.305). Subsequent properties that influenced cluster classification were as follows (Fig. 1*B*): curve midpoint (0.101), membrane capacitance (0.074), rheobase (0.070), input resistance (0.066), threshold (0.064) and sag amplitude (0.053). To investigate how each membrane feature shapes cluster distribution we dimensionally reduced the dataset using the PCA tool. The vector factor map (Fig. 1*C*) showed the contribution of each feature to the first two components (PCs), where PC1 and PC2 accounted for 24.40% and 16.83% of the variance in the dataset, respectively. We found that assignment to type 1 cluster was mainly driven by amplitude change, mean and maximum spiking frequency (Fig. 1*C*). Type 2 was determined by rheobase, membrane capacitance and the midpoint of the F-I curve (Fig. 1*C*), whereas steepness of the F-I curve, input resistance, sag amplitude as well as AP half-width influenced the distribution of type 3. Then we checked whether these electrophysiological features vary across types. Indeed, some of the membrane properties differ significantly between clusters (Fig. 2).

As a result of hierarchical clustering, L2/3 VIP-INs were assigned to three clusters: type 1 (68%), type 2 (12%) and type 3 (20%) (Fig. 1*D*). In respect to Petilla terminology, the vast majority of type 1 VIP-INs resembled cells with

continuously adapting firing, type 2 usually displayed a burst firing, whilst type 3 had an irregular spiking pattern (Fig. 1*D*) (Ascoli et al., 2008).

Next, we asked how GABAbRs modulate activity of these different VIP-IN types. GABAbR activation was assessed by bath application of the GABAbR agonist baclofen (Bac). In agreement with canonical inhibitory effects of GABAbR activation, baclofen administration led to reduction of $R_{in}$ and moderate hyperpolarization of $V_m$ in all three types (Table 1). We further checked how baclofen impacts intrinsic excitability of VIP-INs depending on the cluster type of these interneurons. Here, we examined effects of the GABAbR agonist on the F-I curve using raw data (Fig. 3*A*, *B*, *E*, *F*, *I* and *J*). Also, we employed the analysis of parameters of the sigmoidal Hill function fitted to the F-I curve (Fig. 3*C*, *G* and *K*), due to a vast diversity in F-I curves within the same electrophysiological type. Unexpectedly, both analyses consistently showed that intrinsic excitability of type 1 increased after application of the GABAbR agonist (Fig. 3*A*–*D*), whereas types 2 and 3 were insensitive to this modulation (Fig. 3*E*–*L*). The F-I curve of raw data in type 1 showed increased firing frequency after baclofen application at several current intensity steps (Fig. 3*B*, $P < 0.05$, paired *t* test). Type 1 responded with increased maximum frequency by 19% after baclofen in comparison to control condition (Fig. 3*D*, $41.82 \pm 21.32$ Hz in Ctrl *vs.* $49.58 \pm 25.17$ Hz in Bac, $P = 0.022$, $n = 34$, paired *t* test) and decreased rheobase (Fig. 3*D*, $59.41 \pm 21.95$ pA in Ctrl vs. $51.76 \pm 22.94$ pA in Bac, $P = 0.001$, $n = 34$, Wilcoxon test). There were no significant changes in the steepness of the curve (data not shown, $3.84 \pm 1.27$ Hz/pA in Ctrl *vs.* $3.79 \pm 1.44$ Hz/pA in Bac, $P = 0.856$, $n = 34$, paired *t* test) or midpoint (data not shown, $112.58 \pm 64.55$ pA in Ctrl *vs.* $102.09 \pm 42.70$ pA in Bac, $P = 0.270$, $n = 34$, Wilcoxon test). In type 2, there was no significant change in firing frequency in the raw F-I curve in a stepwise comparison after baclofen (Fig. 3*E* and *F*, $P < 0.05$,

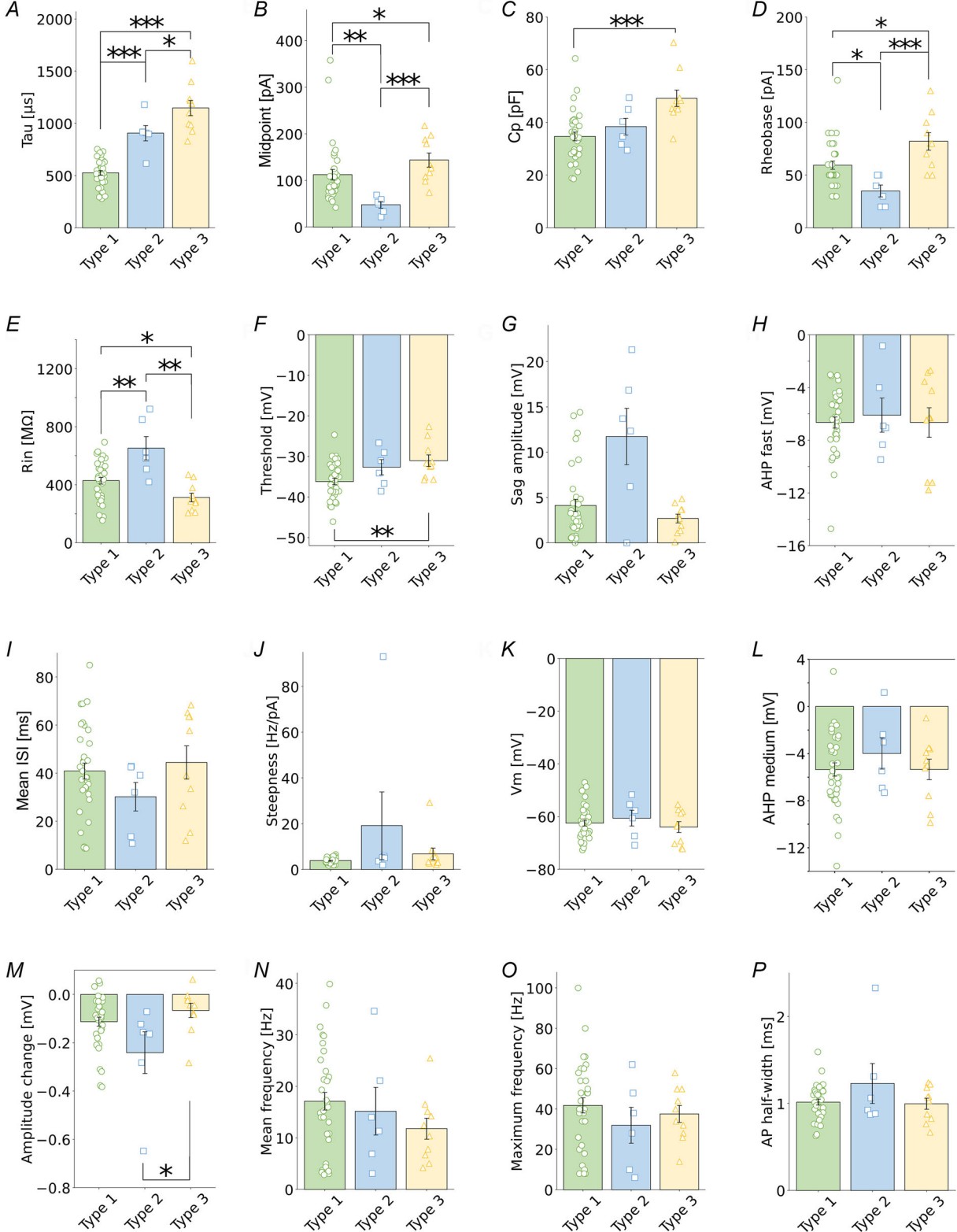

**Figure 2. Basic electrophysiological features of L2/3 VIP-INs grouped into three types using hierarchical clustering**

Data presented as mean (±SD) and individual data points of all 3 VIP-IN types. Graphs are ordered in accordance with feature importance score. *A*, tau (type 1 *vs.* type 2, *P* < 0.001; type 2 *vs.* type 3, *P* = 0.047; type 1 *vs.* type 3,

$P < 0.001$, one-way ANOVA test with Holm–Šidák *post hoc* test). B, midpoint (type 1 *vs.* type 2, $P = 0.002$; type 2 *vs.* type 3, $P < 0.001$; type 1 *vs.* type 3, $P = 0.035$, Kruskal–Wallis test with Dunn's *post hoc* test). C, membrane capacitance ($Cp$) (type 1 *vs.* type 2, $P = 0.374$; type 2 *vs.* type 3, $P = 0.079$; type 1 *vs.* type 3, $P < 0.001$, one-way ANOVA test with Holm–Šidák *post hoc* test). D, rheobase (type 1 *vs.* type 2 $P = 0.026$; type 2 *vs.* type 3 $P < 0.001$; type 1 *vs.* type 3 $P = 0.026$, Kruskal–Wallis test with Dunn's *post hoc* test). E, input resistance ($R_{in}$) (type 1 *vs.* type 2, $P = 0.002$; type 2 *vs.* type 3, $P = 0.001$; type 1 *vs.* type 3, $P = 0.015$, one-way ANOVA test with Holm–Šidák *post hoc* test). F, threshold (type 1 *vs.* type 2, $P = 0.171$; type 2 *vs.* type 3, $P = 0.499$; type 1 *vs.* type 3, $P = 0.009$, one-way ANOVA test with Holm–Šidák *post hoc* test). G, sag amplitude ($P = 0.060$, Kruskal–Wallis test). H, fast afterhyperpolarization (AHP) ($P = 0.901$, one-way ANOVA test). I, mean interevent interval (ISI) ($P = 0.349$, one-way ANOVA test). J, steepness ($P = 0.386$, Kruskal–Wallis test). K, membrane potential ($V_m$) ($P = 0.614$, one-way ANOVA test). L, medium afterhyperpolarization (AHP) ($P = 0.615$, one-way ANOVA test). M, amplitude change (type 1 *vs.* type 2, $P = 0.182$; type 2 *vs.* type 3, $P = 0.038$; type 1 *vs.* type 3, $P = 0.182$, Kruskal–Wallis test with Dunn's *post hoc* test). N, mean frequency ($P = 0.304$, one-way ANOVA test). O, maximum frequency ($P = 0.517$, one-way ANOVA test). P, action potential (AP) half-width ($P = 0.917$, Kruskal–Wallis test). \*$P < 0.05$, \*\*$P < 0.01$, \*\*\*$P < 0.001$. [Colour figure can be viewed at wileyonlinelibrary.com]

paired *t* test), or in maximum frequency (Fig. 3*G* and *H*, $32.00 \pm 21.76$ Hz in Ctrl *vs.* $36.00 \pm 20.04$ Hz in Bac, $P = 0.296$, $n = 6$, paired *t* test), or in rheobase (Fig. 3*H*, $35.00 \pm 13.78$ pA in Ctrl *vs.* $35.00 \pm 10.48$ pA in Bac, $P = 1.00$, $n = 6$, paired *t* test). There was also no effect on either steepness (data not shown, $19.14 \pm 33.08$ Hz/pA in Ctrl *vs.* $4.37 \pm 0.94$ Hz/pA in Bac, $P = 0.688$, $n = 6$, Wilcoxon test) or midpoint of the fitted curve after baclofen in type 2 (data not shown, $47.94 \pm 17.30$ pA in Ctrl *vs.* $50.50 \pm 9.53$ pA in Bac, $P = 0.756$, $n = 6$, paired *t* test). Lastly, the F-I curve of type 3 VIP-INs did not change statistically after baclofen application (Fig. 3*I* and *J*, $P < 0.05$, paired *t* test). Also, there was no change in maximum frequency (Fig. 3*K* and *L*, $37.60 \pm 13.32$ Hz in Ctrl *vs.* $38.8 \pm 17.28$ Hz in Bac, $P = 0.798$, $n = 10$, paired *t* test) or rheobase (Fig. 3*L*, $82.00 \pm 26.16$ pA in Ctrl *vs.* $87.00 \pm 33.35$ pA in Bac, $P = 0.413$, $n = 10$, paired *t* test). Neither of the curve parameters changed (data not shown, steepness: $6.81 \pm 7.67$ Hz/pA in Ctrl *vs.* $5.98 \pm 4.16$ Hz/pA in Bac, $P = 0.922$, $n = 10$, Wilcoxon test; midpoint: $143.95 \pm 47.88$ pA in Ctrl *vs.* $150.01 \pm 54.80$ pA in Bac, $P = 0.449$, $n = 10$, paired *t* test).

Summarizing, in principle GABAbR activation changes membrane properties in a way that should lead to a decrease of neuronal excitability (Table 1). However, the inhibitory effect is minimal and unexpectedly GABAbRs increase intrinsic excitability of the majority of L2/3 VIP-IN (type 1, comprising 68% of the recorded population). Moreover, intrinsic excitability of a subset of VIP-INs (types 2 and 3) is immune to GABAbR modulation. Here, we performed our recordings in regular ACSF (rACSF) because such a recording condition is a standard procedure in the field of whole-cell patch-clamp technique in acute brain slices. For this reason, we could compare our results to the literature. At this stage of our experiments, we speculated that the excitatory effect of GABAbR activity is caused by the fact that rACSF has a higher $Ca^{2+}$ concentration (2.5 mM) in comparison to physiological CSF (Somjen, 2004), which contains about 1 mM.

## Postsynaptic GABAbRs increase VIP-IN intrinsic excitability through a reduction of BK channel activity

Since baclofen administration increased the intrinsic excitability of type 1, in subsequent experiments we aimed to unravel the mechanisms of this phenomenon. We hypothesized that in the condition of a high extracellular $Ca^{2+}$ level in rACSF, GABAbR activation might lead to indirect inhibition of BK channels in L2/3 VIP-INs through suppression of the N-type calcium channel activity, as it has been previously described in rat retinal ganglion cells (Garaycochea & Slaughter, 2016) and calyx terminals in semicircular canal cristae (Ramakrishna & Sadeghi, 2020).

To test this hypothesis, we analysed how the F-I curve changes after bath application of the BK channel blocker (paxilline, Pax) (Fig. 4). As before, VIP-INs were hierarchically clustered based on their 18 electrophysiological properties. However, here and for the following experiments, we pooled types 2 and 3 into one cluster, because these types belong to the same dendrogram clade (Fig. 1*A*) and responded similarly to GABAbR activation with baclofen (Fig. 3*E–L*). In general, we observed that paxilline increased the intrinsic excitability of all three types (Fig. 4). Across all types, there were pronounced changes in the mean raw F-I curve in the stepwise comparison, especially in low current intensities but not in high current intensities when cells reached maximal frequencies (Fig. 4*B* and *F*, $P < 0.05$, paired *t* test). Nonetheless, paxilline reduced rheobase by 28% in type 1 (Fig. 4*D*, $40.83 \pm 10.37$ pA in Ctrl *vs.* $29.16 \pm 11.14$ pA in Pax, $P = 0.002$, $n = 12$, Wilcoxon test) and by 34% in type 2 + 3 (Fig. 4*H*, $87.00 \pm 24. 06$ pA in Ctrl *vs.* $57.00 \pm 22.63$ pA in Pax, $P < 0.001$, $n = 10$, paired *t* test). Moreover, in type 1, the analysis of the fitted curves showed a significant decrease in the midpoint of a curve by 30% after Pax (data not shown, $107.08 \pm 44.34$ pA in Ctrl *vs.* $74.46 \pm 24.70$ pA in Pax, $P = 0.002$, $n = 12$, Wilcoxon test). No changes were found in maximum frequency (Fig. 4*C* and *D*, $56.83 \pm 23.62$ Hz in Ctrl *vs.* $66.00 \pm 30.80$ Hz in Pax, $P = 0.177$, $n = 12$, paired *t* test), or steepness of a curve (data not shown, $2.80 \pm 1.00$ Hz/pA

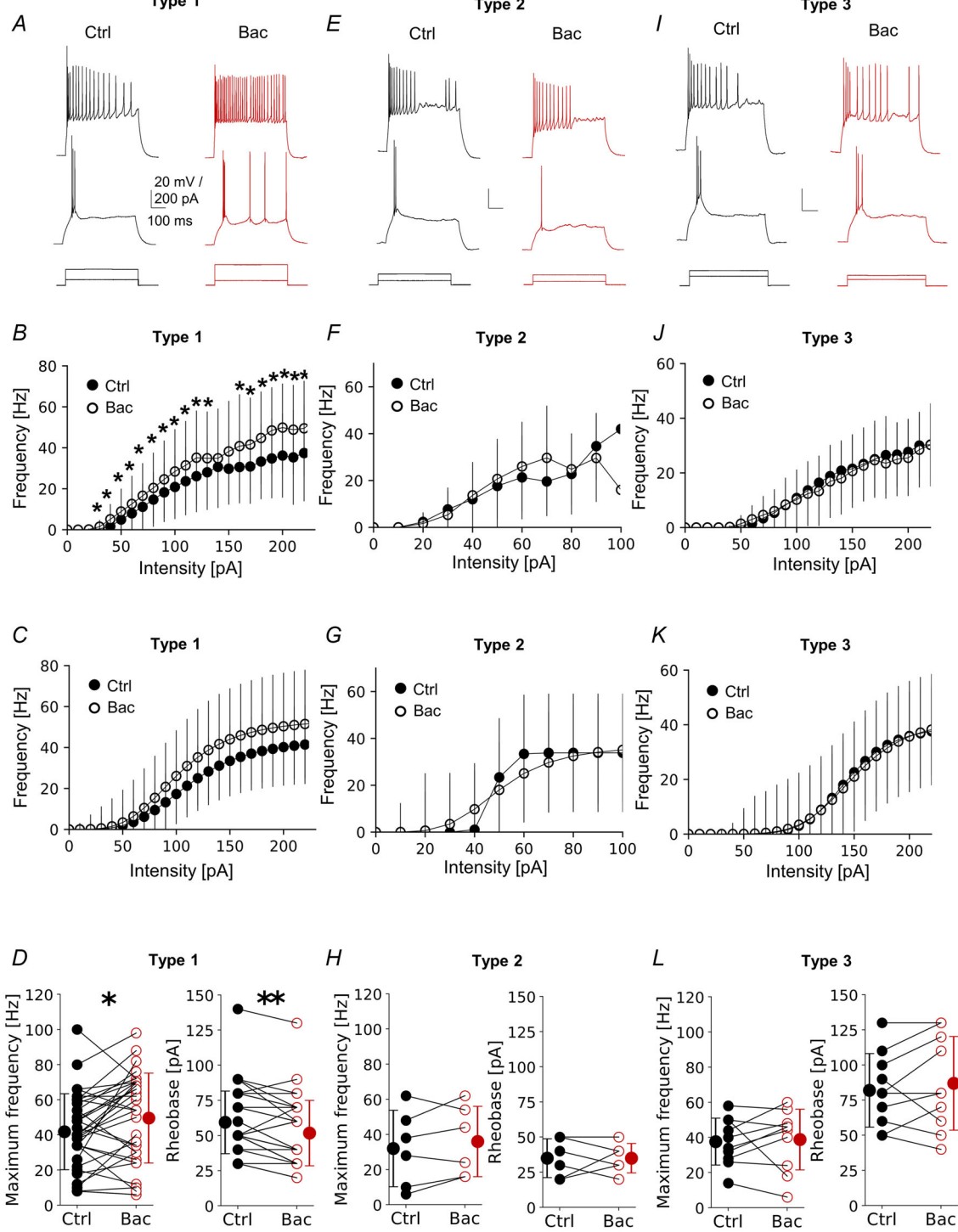

**Figure 3. GABAbRs increase L2/3 VIP-IN intrinsic excitability in an interneuron type-dependent manner**

*A*, example traces of type 1 VIP-IN firing responses upon 500 ms-long depolarizing current injection. Lower traces show step intensities of current injected into the soma, middle traces show neuronal firings at rheobase, upper traces represent maximum spike frequency before (Ctrl, left) and after the GABAbR agonist (baclofen, Bac, right). *B*, mean (±SD) firing frequency–current intensity (F-I) curves for type 1 VIP-INs in control ACSF and after baclofen ($P < 0.05$, paired *t* test, *n* = 34). *C*, Hill sigmoidal curves fitted to the data shown in *B*, *n* = 34. *D*, left, within-cell comparison and mean (±SD) maximum firing frequency of type 1 in control and after baclofen ($P = 0.022$, paired *t* test, *n* = 34). Right, same but for rheobase ($P = 0.001$, Wilcoxon test, *n* = 34). *E–H*, same as *A–D* but for type 2 VIP-INs ($P = 0.296$ and $P = 1.000$, respectively paired *t* test, *n* = 6). *I–L*, same as *A–D* but for type 3 VIP-INs ($P = 0.798$ and $P = 0.413$, respectively, paired *t* test, *n* = 10). *$P < 0.05$, **$P < 0.01$. [Colour figure can be viewed at wileyonlinelibrary.com]

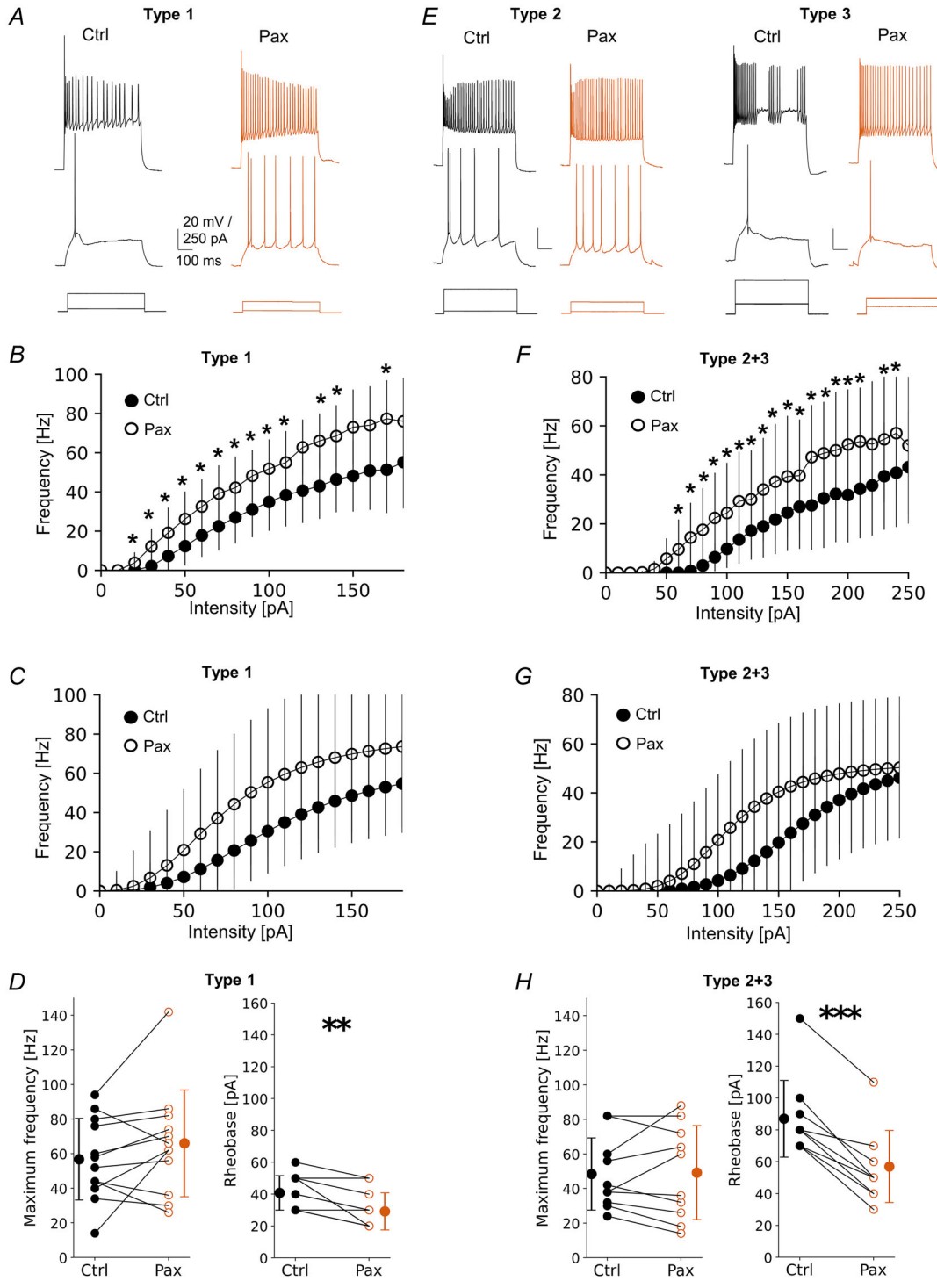

**Figure 4. BK channels suppress L2/3 VIP-IN intrinsic excitability at high extracellular Ca²⁺ level**

*A*, examples of firing responses of type 1 upon a 500 ms-long pulse. Lower traces show step intensities of current injected to the soma, middle traces show neuronal firings at rheobase, upper traces represent maximum spike frequency in control ACSF (Ctrl, left) and after application of the BK channel antagonist (paxilline, Pax, right). *B*, mean (±SD) firing frequency–current curves type 1 in control and after Pax ($P < 0.05$, paired *t* test, $n = 12$). *C*, Hill sigmoidal curves fitted to the data shown in *B*, $n = 12$. *D*, left, within-cell comparison and mean (±SD) maximum firing frequency of type 1 in control and after Pax ($P = 0.177$, paired *t* test, $n = 12$). Right, same but for rheobase ($P = 0.002$, Wilcoxon test, $n = 12$). *E–H*, same as *A–D* but for pooled raw data of types 2 and 3 (type 2 + 3) ($P = 0.820$ Wilcoxon test and $P < 0.001$ paired *t* test, respectively, $n = 10$). *$P < 0.05$, **$P < 0.01$, ***$P < 0.001$. [Colour figure can be viewed at wileyonlinelibrary.com]

in Ctrl *vs.* 2.65 ± 1.32 Hz/pA in Pax, $P = 0.511$, $n = 12$, paired *t* test). Similarly, in type 2 + 3 there was a significant reduction in the midpoint after Pax (data not shown, 167.66 ± 47.15 pA in Ctrl *vs.* 110.58 ± 30.32 pA in Pax, $P = 0.003$, $n = 10$, paired *t* test). Also, no changes in the maximum frequency were noted (Fig. 4*G* and *H*, 48.40 ± 19.77 Hz in Ctrl *vs.* 49.20 ± 25.80 Hz in Pax, $P = 0.820$, $n = 10$, Wilcoxon test) nor in steepness of a curve (data not shown, 4.73 ± 1.99 Hz/pA in Ctrl *vs.* 4.06 ± 1.93 Hz/pA in Pax, $P = 0.383$, $n = 10$, paired *t* test).

These results show that the blockade of BK channels shifted the F-I curve to the left indicating that these channels downregulate intrinsic excitability of L2/3 VIP-INs. Summarizing, BK channels reduce the intrinsic excitability of L2/3 VIP-INs regardless of their electrophysiological type.

Subsequently, we asked whether the effect of GABAbR activation will be occluded by the previous blockade of BK channels. For this purpose, brain slices were perfused with paxilline followed by paxilline with baclofen (Fig. 5). In the raw data, we observed statistically significant changes in the F-I curves, but only in low current intensities in type 1 (Fig. 5*A* and *B*, $P < 0.05$, paired *t* test), which was also reflected in reduced rheobase (Fig. 5*D*, 50.00 ± 22.11 Hz in Pax *vs.* 42.00 ± 26.16 Hz in Pax + Bac, $P = 0.02$, $n = 10$, paired *t* test). There was no effect of baclofen in the presence of paxilline in type 2 + 3 (Fig. 5*E* and *F*, $P < 0.05$, paired *t* test). However, analysis of the fitted curves showed no differences between both conditions in all types. When BK channels were blocked with paxilline, GABAbR agonist application had no effect on the maximum frequency (Fig. 5*C* and *D*, 50.60 ± 12.18 Hz in Pax *vs.* 51.00 ± 14.97 Hz in Pax + Bac, $P = 0.823$, $n = 10$, paired *t* test), steepness (data not shown, 3.78 ± 1.23 Hz/pA in Pax *vs.* 3.06 ± 0.93 Hz/pA in Pax + Bac, $P = 0.084$, $n = 10$, Wilcoxon test) or midpoint in type 1 (data not shown, 99.72 ± 48.42 pA in Pax *vs.* 96.35 ± 65.20 pA in Pax + Bac, $P = 0.160$, $n = 10$, Wilcoxon test). Also, there were no differences in maximum frequency (Fig. 5*G* and *H*, 42.57 ± 14.54 Hz in Pax *vs.* 43.42 ± 17.38 Hz in Pax + Bac, $P = 0.802$, $n = 7$, paired *t* test), rheobase (Fig. 5*H*, 58.57 ± 18.07 pA in Pax *vs.* 61.42 ± 35.22 pA in Pax + Bac, $P = 0.875$, $n = 7$, Wilcoxon test), steepness (data not shown, 5.01 ± 3.46 Hz/pA in Pax *vs.* 4.12 ± 2.06 Hz/pA in Pax + Bac, $P = 0.481$, $n = 7$, paired *t* test) or midpoint of the curve in type 2 + 3 (data not shown, 111.62 ± 48.96 pA in Pax *vs.* 118.38 ± 78.17 pA in Pax + Bac, $P = 0.682$, $n = 7$, paired *t* test).

Thus, the effect of GABAbR activation on L2/3 VIP-IN intrinsic excitability was partially occluded by the previous inactivation of BK channels in the VIP-IN type that was sensitive to GABAbR activation (type 1). These results suggest that postsynaptic GABAbRs inhibit the activity of BK channels, leading to an increase in intrinsic excitability of a subset of L2/3 VIP-INs.

## Presynaptic GABAbRs increase VIP-IN intrinsic excitability via a reduction of GABAaR activity

In the presence of Pax, baclofen still slightly enhanced the intrinsic excitability of type 1 (Fig. 5), indicating an additional mechanism of GABAbR action. Here, we hypothesized that presynaptic GABAbRs might suppress GABA release, which in turn reduces an inhibitory effect of GABAaRs on intrinsic excitability. We have previously found that L2/3 VIP-IN intrinsic excitability is decreased through GABAaRs, probably due to a high level of tonic GABAaR-mediated inhibition (Bogaj et al., 2023). To unfold the remaining mechanism of intrinsic excitability enhancement in type 1 VIP-INs, we analysed the F-I curve in the condition when GABAaRs were blocked with picrotoxin (PTX) (Fig. 6). Now, when both GABAaRs and BK channels were blocked with PTX and Pax, respectively, a subsequent application of the GABAbR agonist had in general no effects on the F-I curves in all three types of VIP-INs. No changes were observed after baclofen application in the raw F-I curves for either type 1 (Fig. 6*A* and *B*, $P < 0.05$, paired *t* test) or type 2 + 3 (Fig. 6*E* and *F*, $P < 0.05$, paired *t* test). Furthermore, curve fitting also did not reveal any significant changes between the two conditions. For type 1, baclofen administration did not influence maximum frequency (Fig. 6*C* and *D*, 85.60 ± 30.01 Hz in PTX + Pax *vs.* 86.40 ± 49.18 Hz in PTX + Pax + Bac, $P = 0.933$, $n = 5$, paired *t* test), steepness (data not shown, 2.22 ± 0.69 Hz/pA in PTX + Pax *vs.* 2.31 ± 0.70 Hz/pA in PTX + Pax + Bac, $P = 0.682$, $n = 5$, paired *t* test), midpoint (data not shown, 99.20 ± 68.68 pA in PTX + Pax *vs.* 81.00 ± 48.29 pA in PTX + Pax + Bac, $P = 0.162$, $n = 5$, paired *t* test), or rheobase (Fig. 6*C* and *D*, 32.00 ± 4.00 pA in PTX + Pax *vs.* 28.00 ± 7.48 pA in PTX + Pax + Bac, $P = 0.500$, $n = 5$, Wilcoxon test). Likewise, type 2 + 3 did not respond to GABAbR activation, and curve parameters such as maximum frequency (Fig. 6*G* and *H*, 68.00 ± 22.05 Hz in PTX + Pax *vs.* 66.00 ± 29.82 Hz in PTX + Pax + Bac, $P = 0.749$, $n = 6$, paired *t* test), steepness (data not shown, 2.99 ± 1.29 Hz/pA in PTX + Pax *vs.* 3.49 ± 1.64 Hz/pA in PTX + Pax + Bac, $P = 0.506$, $n = 6$, paired *t* test), midpoint (data not shown, 214.81 ± 112.05 pA in PTX + Pax *vs.* 183.10 ± 54.60 pA in PTX + Pax + Bac, $P = 1.00$, $n = 6$, Wilcoxon test) and rheobase (Fig. 6*G* and *H*, 65.00 ± 13.78 pA in PTX + Pax *vs.* 66.66 ± 12.11 pA in PTX + Pax + Bac, $P = 0.611$, $n = 6$, paired *t* test) were not statistically different.

Altogether, our results indicate that GABAbRs increase the intrinsic excitability of L2/3 VIP-INs through at least two mechanisms. Firstly, postsynaptic GABAbRs inhibit BK channels (Fig. 5) and secondly, presynaptic GABAbRs suppress GABA release and thus diminish the activity of postsynaptic GABAaRs (Fig. 6). The canonical inhibitory effect of GABAbRs on neuronal activity is probably outweighed by these two excitatory mechanisms.

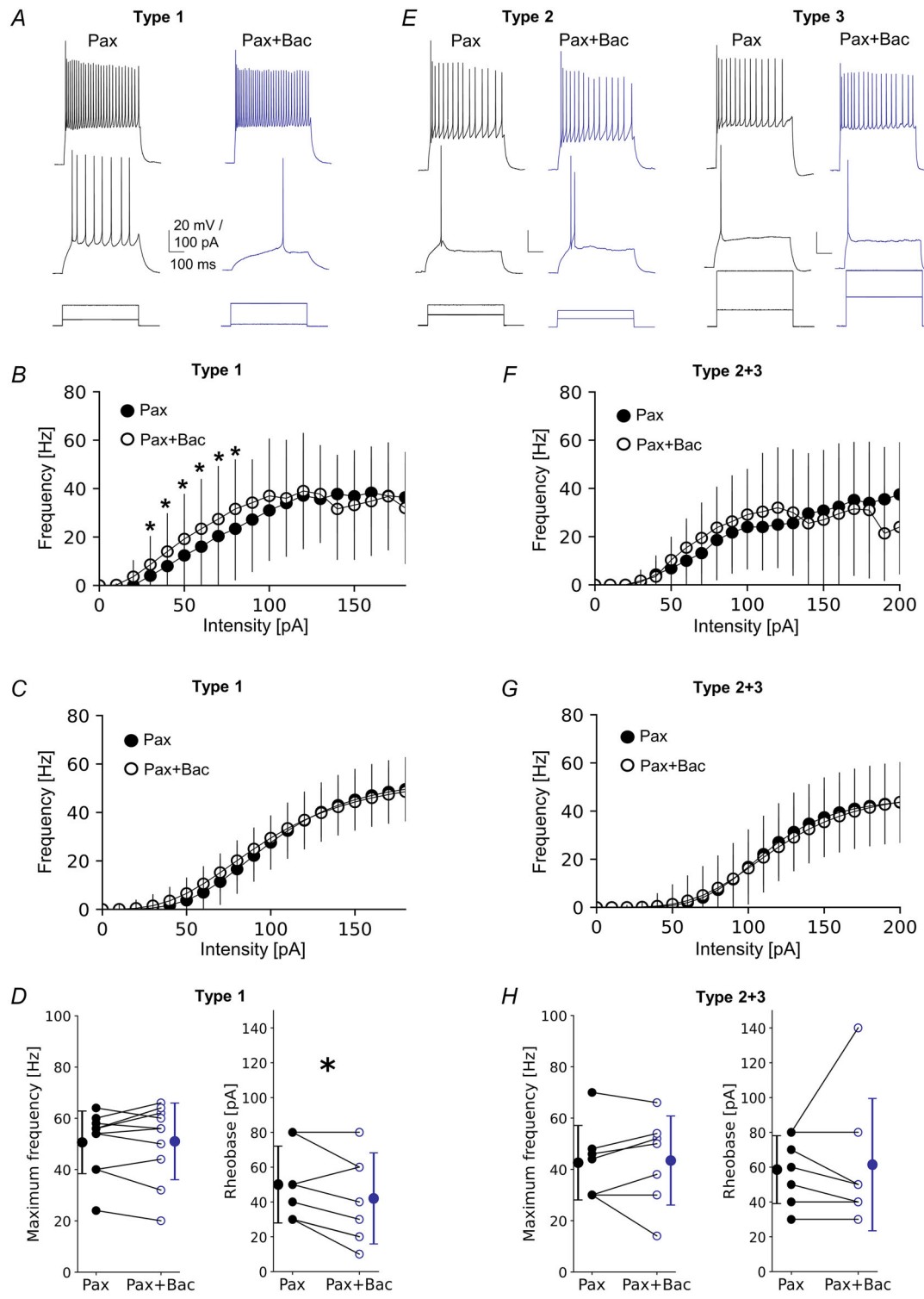

**Figure 5. Pharmacological blockade of BK channels partially occludes an excitatory effect of GABAbRs on L2/3 VIP-IN intrinsic excitability**

*A*, examples of firing responses of type 1 upon a 500 ms-long current step. Lower traces show current intensities injected to the soma, middle traces show neuronal firings at rheobase, upper traces represent maximum spike frequency in ACSF with the BK channel antagonist (Pax, left) and in ACSF with the BK channel antagonist and GABAbR agonist (Pax + Bac, right). *B*, mean (±SD) firing frequency–current intensity curves for type 1 in Pax and after Pax + Bac (*P* < 0.05, paired *t* test, *n* = 10). *C*, Hill sigmoidal curves fitted to the data shown in *B*, *n* = 10. *D*, left, within-cell comparison and mean (±SD) maximum firing frequency of type 1 in ACSF with Pax and after

Pax + Bac (*P* = 0.823, paired *t* test, *n* = 10). Right, same but for rheobase (*P* = 0.022, Wilcoxon test, *n* = 10). *E–H*, same as *A–D* but for pooled raw data of types 2 and 3 (*P* = 0.802 paired *t* test and *P* = 0.875 Wilcoxon test, respectively, *n* = 7). *P* < 0.05. [Colour figure can be viewed at wileyonlinelibrary.com]

**Table 2. Membrane properties before and after the GABAbR agonist (baclofen) application in mACSF**

| VIP-IN type | Membrane potential (mV) | | | Input resistance (MΩ) | | |
|---|---|---|---|---|---|---|
| | Ctrl | Bac | *P*-value (*n* cells) | Ctrl | Bac | *P*-value (*n* cells) |
| Type 1 | −52.73 ± 6.39 | −55.97 ± 7.82 | 0.183 (8) paired *t* test | 434.50 ± 93.22 | 374.71 ± 108.03 | 0.039* (8) Wilcoxon test |
| Type 2 + 3 | −53.78 ± 8.81 | −55.90 ± 7.82 | 0.061 (10) paired *t* test | 260.09 ± 98.86 | 231.74 ± 75.16 | 0.072 (10) paired *t* test |

Data are means ± SD. *Statistical significance.

## Canonical inhibitory effect of GABAbRs arises in physiological extracellular Ca$^{2+}$ level

The previous experiments (Figs 1–6) were performed in rACSF when the neuronal network is silent and a relatively high level of extracellular Ca$^{2+}$ (2.5 mM) promotes all the mechanisms that depend on the Ca$^{2+}$ concentration (Borst, 2010; Urban-Ciecko et al., 2015, 2018). For this reason, we wanted to know whether the excitatory effects of GABAbRs on L2/3 VIP-IN activity would be maintained in more physiological conditions when the neuronal network is spontaneously active (Kanigowski et al., 2023; Maffei et al., 2004; Sanchez-Vives & McCormick, 2000; Urban-Ciecko et al., 2015, 2018). Here, we performed recordings in modified ACSF (mACSF), containing a lower (1 mM) and thus more physiological concentration of Ca$^{2+}$ (Maffei et al., 2004; Sanchez-Vives & McCormick, 2000). First, we compared membrane properties before and after baclofen application in this condition (Table 2). As in rACSF (Table 1), baclofen slightly hyperpolarized $V_m$ and reduced $R_{in}$ in all VIP-IN types in mACSF (Table 2). In mACSF, as for rACSF (Fig. 4), the intrinsic excitability of type 1 VIP-INs changed after GABAbR agonist application (Fig. 7*A–D*), while type 2 + 3 remained insensitive (Fig. 7*E–H*). Interestingly, now baclofen reduced the intrinsic excitability of type 1 VIP-INs in mACSF (Fig. 7*A–D*), as expected from the canonical inhibitory role of GABAbRs. The raw F-I curve of type 1 was reduced in baclofen in comparison to Ctrl in the current steps at higher intensities but not at lower intensities (Fig. 7*A* and *B*). The maximum frequency of type 1 decreased by 20% (Fig. 7*C* and *D*, 95.75 ± 27.59 Hz in Ctrl *vs.* 76.25 ± 27.90 Hz in Bac, *P* = 0.004, *n* = 8, paired *t* test), whereas no differences were observed in steepness (data not shown, 2.48 ± 0.64 Hz/pA in Ctrl *vs.* 2.84 ± 0.95 Hz/pA in Bac, *P* = 0.242, *n* = 8, paired *t* test), the midpoint of the fitted curve (data not shown,

58.94 ± 18.47 pA in Ctrl *vs.* 52.81 ± 25.25 pA in Bac, *P* = 0.298, *n* = 8, paired *t* test) or rheobase (Fig. 7*D*, 25.00 ± 7.07 pA in Ctrl *vs.* 27.50 ± 11.98 pA in Bac, *P* = 0.563, *n* = 8, Wilcoxon test). For type 2 + 3, the raw F-I curve did not differ between conditions indicating that these interneurons are insensitive to GABAbR activation in mACSF (Fig. 7*E–H*) as it was in rACSF (Fig. 3*E–L*). There was no effect of GABAbR activation on type 2 + 3 VIP-INs in the maximum frequency (Fig. 7*G* and *H*, 61.60 ± 9.74 Hz in Ctrl *vs.* 51.80 ± 23.86 Hz in Bac, *P* = 0.121, *n* = 10, paired *t* test), steepness (data not shown, 5.63 ± 2.24 Hz/pA in Ctrl *vs.* 5.35 ± 1.34 Hz/pA in Bac, *P* = 0.697, *n* = 10, paired *t* test), midpoint (data not shown, 85.10 ± 68.21 pA in Ctrl *vs.* 87.76 ± 57.07 pA in Bac, *P* = 0.848, *n* = 10, paired *t* test) or rheobase (Fig. 7*H*, 58.00 ± 32.93 pA in Ctrl *vs.* 61.00 ± 36.34 pA in Bac, *P* = 0.748, *n* = 10, paired *t* test).

We next investigated how BK channels regulate the intrinsic excitability of VIP-INs in mACSF. Paxilline was bath-applied and F-I curves were analysed (Fig. 8). Now, no differences were found in the raw F-I curves after Pax application in any of the three types (Fig. 8*A*, *B*, *E* and *F*). Also, no significant differences in parameters of the sigmoidal functions were observed after paxilline in any of the types (Fig. 8*C*, *D*, *G* and *H*). For type 1 VIP-INs, there were no differences in maximum frequency (Fig. 8*C* and *D*, 77.22 ± 20.18 Hz in Ctrl *vs.* 73.00 ± 20.54 Hz in Pax, *P* = 0.430, *n* = 12, paired *t* test), steepness (data not shown, 3.32 ± 0.82 Hz/pA in Ctrl *vs.* 3.24 ± 1.12 Hz/pA in Pax, *P* = 0.804, *n* = 12, paired *t* test), midpoint (data not shown, 58.60 ± 29.29 pA in Ctrl *vs.* 58.35 ± 37.29 pA in Pax, *P* = 0.967, *n* = 12, paired *t* test) or rheobase (Fig. 8*D*, 31.66 ± 12.13 pA in Ctrl *vs.* 30.83 ± 13.81 pA in Pax, *P* = 0.813, *n* = 12, Wilcoxon test). No differences for type 2 + 3 were observed either (maximum frequency: Fig. 8*G* and *H*, 73.00 ± 12.80 Hz in Ctrl *vs.* 65.50 ± 17.23 Hz in Pax, *P* = 0.06, *n* = 4, paired *t* test; steepness: data not shown, 3.50 ± 0.54 Hz/pA in Ctrl *vs.* 3.58 ± 1.05 Hz/pA

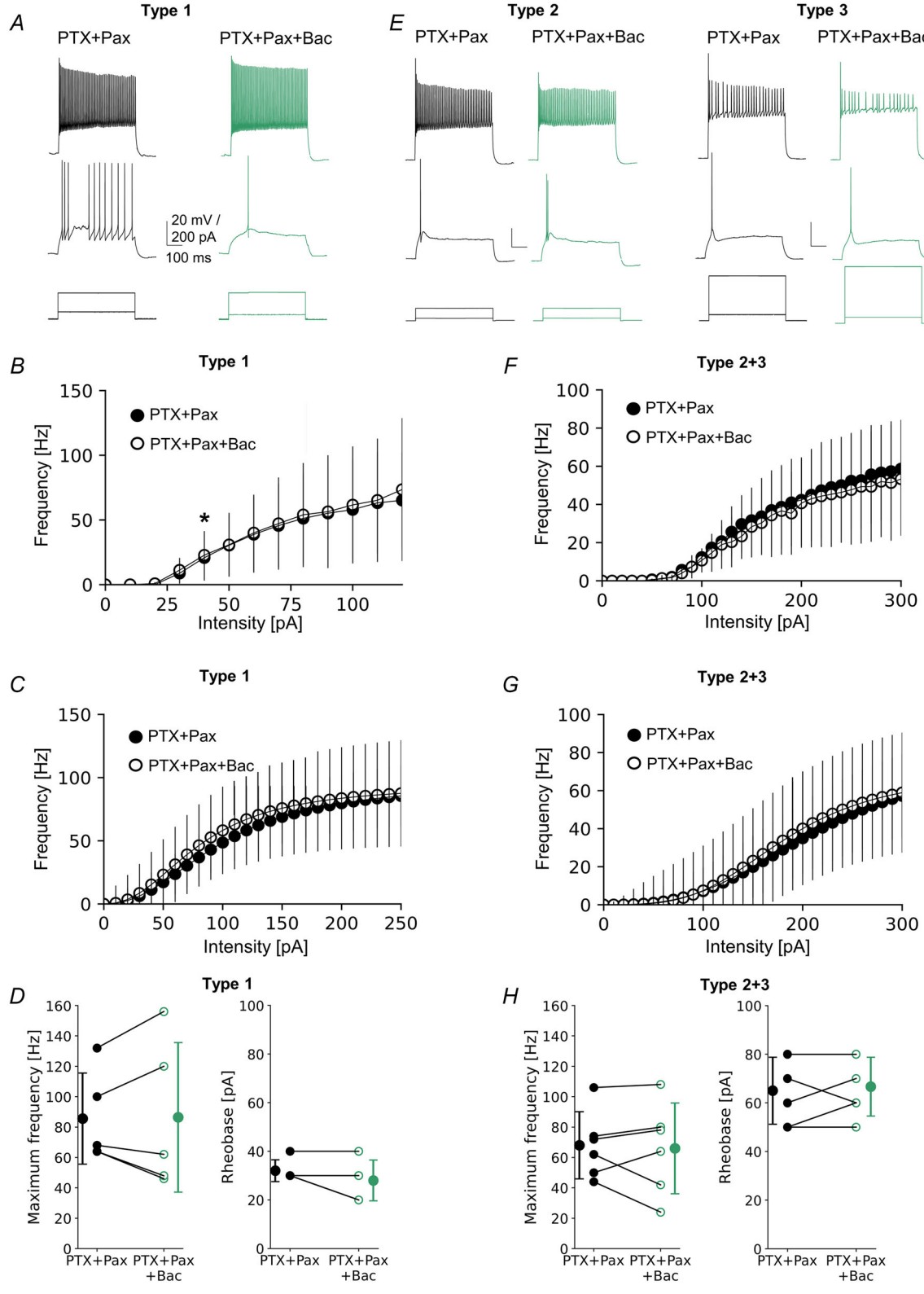

**Figure 6. Presynaptic GABAbRs increase L2/3 VIP-IN intrinsic excitability indirectly, through a reduction of GABAaR-mediated inhibition**

*A*, examples of firing responses of type 1 upon a 500 ms-long pulse. Lower traces show step intensities of current injected to the soma, middle traces show neuronal firings at rheobase, and upper traces represent maximum spike frequency in ACSF with the GABAaR blocker (picrotoxin, PTX) and BK channel antagonist paxilline (PTX + Pax, left) followed by a cocktail composed of picrotoxin, paxilline and baclofen (PTX + Pax + Bac, right). *B*, mean (±SD)

firing frequency–current intensity curves for type 1 in ACSF with PTX + Pax followed by PTX + Pax + Bac ($P < 0.05$, paired *t* test, $n = 5$). *C*, Hill sigmoidal curves fitted to the data shown in *B*, $n = 5$. *D*, left, within-cell comparison and mean (±SD) maximum firing frequency of type 1 in ACSF with PTX + Pax and after PTX + Pax + Bac ($P = 0.933$, paired *t* test, $n = 5$). Right, same but for rheobase ($P = 0.500$, Wilcoxon test, $n = 5$). *E–H*, same as *A–D* but for pooled raw data of type 2 + 3 VIP-INs ($P = 0.749$ and $P = 0.611$ paired *t* test, respectively, $n = 6$). [Colour figure can be viewed at wileyonlinelibrary.com]

in Pax, $P = 0.865$, $n = 4$, paired *t* test; midpoint: data not shown, $88.97 \pm 44.68$ pA in Ctrl *vs.* $81.85 \pm 54.13$ pA in Pax, $P = 0.286$, $n = 4$, paired *t* test; rheobase: Fig. 8*H*, $47.50 \pm 19.20$ pA in Ctrl *vs.* $42.50 \pm 24.87$ pA in Pax, $P = 1.0$, $n = 4$, Wilcoxon test).

These data indicate that BK channels are presumably inactive in mACSF in contrast to rACSF, hence uncovering the inhibitory effect of postsynaptic GABAbRs. Now the activation of GIRK channels (Gassmann & Bettler, 2012; Kulik et al., 2018) through postsynaptic GABAbRs reduces the intrinsic excitability of VIP-INs.

### GABAbRs decrease excitatory synaptic transmission to L2/3 VIP-INs

Next, we analysed the effect of GABAbR activation on the excitatory drive to L2/3 VIP-INs at the level of synaptic transmission. There were no significant differences between VIP-IN types in regards to the frequency or amplitude of spontaneous postsynaptic currents (sEPSCs) (Fig. 9*A* and *B*, $P = 0.988$ and $P = 0.240$, Kruskal–Wallis test). However, baclofen administration decreased significantly the frequency and amplitude of sEPSCs in type 1 (Fig. 9*C*, Ctrl *vs.* Bac, frequency: $2.31 \pm 2.17$ Hz *vs.* $1.44 \pm 1.44$ Hz, $P = 0.002$, $n = 22$, paired *t* test, amplitude: $16.67 \pm 4.53$ pA *vs.* $15.24 \pm 3.88$ pA, $P < 0.001$, $n = 22$, Wilcoxon test). GABAbR agonist affected also the frequency of sEPSCs of type 2 (Fig. 9*D*, $1.58 \pm 0.23$ Hz in Ctrl *vs.* $1.01 \pm 0.61$ Hz in Bac, $P = 0.033$, $n = 6$, paired *t* test) and type 3 VIP-INs (Fig. 9*E*, $1.88 \pm 1.24$ Hz in Ctrl *vs.* $1.07 \pm 0.71$ Hz in Bac, $P = 0.033$, $n = 5$, paired *t* test), with a trend in reduction of the amplitude (Fig. 9*D* and *E*; type 2: $13.54 \pm 1.96$ pA *vs.* $12.30 \pm 1.61$ pA, $P = 0.107$, $n = 6$, paired *t* test; type 3: $15.09 \pm 1.29$ pA *vs.* $13.45 \pm 1.65$ pA, $P = 0.06$, $n = 5$, paired *t* test).

Summarizing, these results show that presumably pre-synaptic GABAbRs reduce the excitatory drive to all VIP-IN types.

### VIP-INs are tonically modulated by GABAbRs during spontaneous network activity

It has been previously established that SST-INs are highly active in mACSF (Kanigowski et al., 2023; Urban-Ciecko et al., 2015, 2018). Here, we observed that L2/3 VIP-INs are also spontaneously active in mACSF (Fig. 10). In general, types 1 and 2 had similar frequencies of spontaneous firing, whereas type 3 had the lowest activity; however, the differences were not statistically significant (Fig. 10*A*, $3.87 \pm 4.49$ Hz for type 1, $3.10 \pm 3.18$ Hz for type 2 and $1.34 \pm 1.57$ Hz for type 3, $P = 0.280$, Kruskal–Wallis test). In fact, type 3 was mostly inactive (Fig. 10*A*). The spontaneous firing can be shaped by several mechanisms including synaptic drive and intrinsic excitability (Kanigowski et al., 2023). To check how excitatory synaptic drive modulates L2/3 VIP-IN spontaneous firing, glutamate receptors (AMPA and NMDA) were blocked using DNQX and APV, respectively (Fig. 10*B* and *C*). A prominent decrease in neuronal activity was observed after the application of glutamate receptor antagonists. The frequency of spontaneous action potentials (sAPs) decreased in type 1 (Fig. 10*C*, $3.53 \pm 3.34$ Hz in Ctrl *vs.* $1.75 \pm 1.86$ Hz in APV + DNQX, $P = 0.031$, $n = 8$, Wilcoxon test). For type 2 + 3, sAP frequency was unchanged after drug application (Fig. 10*D*, $1.43 \pm 2.02$ Hz in Ctrl *vs.* $0.40 \pm 0.56$ Hz in APV + DNQX, $P = 0.50$, $n = 3$, Wilcoxon test). In the presence of APV + DNQX, we also observed that sAP frequency was different between VIP-IN types (Fig. 10*B*, $6.48 \pm 3.16$ Hz in type 1 *vs.* $2.52 \pm 3.31$ Hz in type 2, $P = 0.022$, two sample *t* test).

Spontaneous firing can be modulated by several mechanisms including those that are dependent on GABAbRs. For this reason, we wanted to know how GABAbRs regulate the spontaneous activity of L2/3 VIP-INs. We found that the frequency of spontaneous firing was reduced by 49% after baclofen application in type 1 (Fig. 10*E*, $4.74 \pm 6.77$ Hz in Ctrl *vs.* $2.38 \pm 3.96$ Hz in Bac, $P = 0.043$, $n = 7$, Wilcoxon test). In contrast, baclofen had no effect on sAPs in type 2 + 3 (Fig. 10*F*, $2.34 \pm 3.10$ Hz in Ctrl *vs.* $3.73 \pm 5.82$ Hz in Bac, $P = 0.945$, $n = 9$, Wilcoxon test). The effect of baclofen on spontaneous activity might be very complex and depend on many pre- and post-synaptic mechanisms including changes of excitatory synaptic drive and intrinsic excitability. For this reason, we wanted to know how the spontaneous activity of VIP-INs is modulated through GABAbRs when glutamate synaptic drive is blocked pharmacologically. Application of the GABAbR agonist in the presence of glutamate receptor antagonists further diminished spontaneous activity of type 1 VIP-INs (Fig. 10*G*, $6.48 \pm 3.16$ Hz in APV + DNQX *vs.* $2.52 \pm 3.31$ Hz in APV + DNQX + Bac, $P = 0.038$,

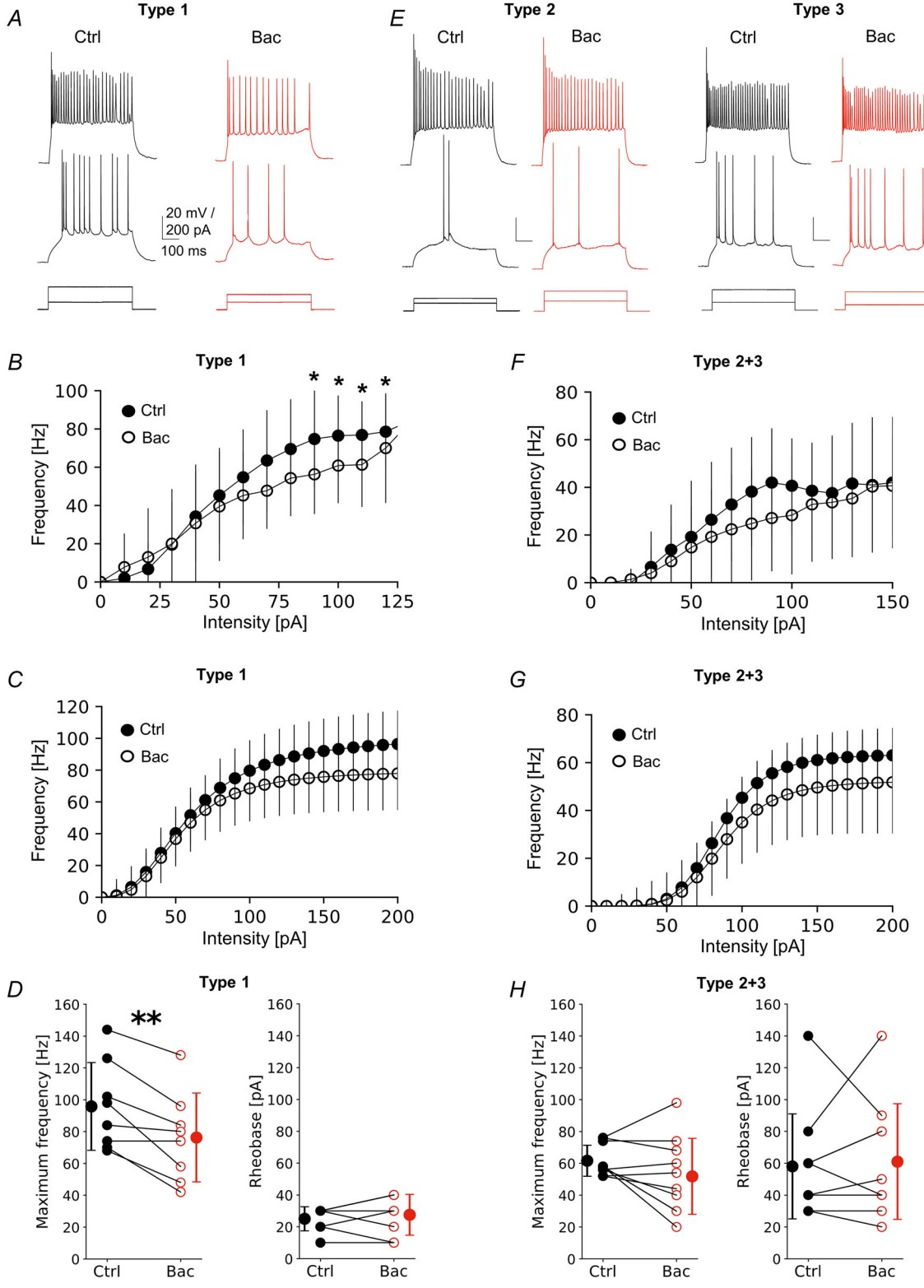

**Figure 7. GABAbRs reduce L2/3 VIP-IN intrinsic excitability at physiological extracellular Ca²⁺ level**

*A*, examples of firing responses of type 1 upon a 500 ms-long pulse. Lower traces show step intensities of current injected to the soma, middle traces show neuronal firings at rheobase, and upper traces represent maximum spike frequency in control ACSF (Ctrl, left) and after bath application of the GABAbR agonist (Bac, right). *B*, mean (±SD) firing frequency–current intensity curves for type 1 in control and after Bac ($P < 0.05$, $n = 8$, paired *t*

*n* = 6, paired *t* test); however, firing frequency of type 2 + 3 remained unchanged (Fig. 10*H*, 2.94 ± 2.62 Hz in APV + DNQX *vs.* 3.07 ± 4.25 Hz in APV + DNQX + Bac, *P* = 0.886, *n* = 7, paired *t* test).

Altogether, these results further confirm our previous results that GABAbRs control the activity of VIP-IN types in distinct manners – type 1 VIP-INs are controlled presynaptically through a reduction of excitatory synaptic drive to these interneurons (Fig. 9*C*) as well as postsynaptically by decreasing their intrinsic excitability (Fig. 3*A–D*), whereas types 2 and 3 are modulated only presynaptically (Figs. 9*D*, *E*).

Additionally, we checked the activity of type 1 and type 2 + 3 VIP-INs after application of the BK channel antagonist. Paxilline did not change spontaneous firing of all types of VIP-INs (type 1: Fig. 10*I*, 3.38 ± 2.03 Hz in Ctrl *vs.* 3.72 ± 5.07 Hz in Pax, *P* = 0.810, *n* = 7, paired *t* test; type 2 + 3: Fig. 10*J*, 2.92 ± 2.01 Hz in Ctrl *vs.* 1.71 ± 2.20 Hz in Pax, *P* = 0.358, *n* = 3, paired *t* test), further confirming that BK channels are inactive at a physiological $Ca^{2+}$ level or their activity has no effect on spontaneous firing.

## Discussion

Here, we show that GABAbRs regulate the activity of L2/3 VIP-INs in two opposite directions depending on recording conditions mainly differing in extracellular $Ca^{2+}$ concentrations. We observed that in standard patch-clamp recording conditions, when the $Ca^{2+}$ concentration is relatively high (2.5 mM), GABAbRs presumably suppress the activity of BK channels and diminish GABAaR-mediated inhibition, leading to an increase in intrinsic excitability of a subclass of L2/3 VIP-INs. When ACSF contains more physiological (low) $Ca^{2+}$ concentration (1 mM), GABAbRs canonically inhibit the activity of these interneurons. Thus, GABAbRs might act as an enhancer or a suppressor of L2/3 VIP-INs depending on the network conditions. Finally, we found that a subset of L2/3 VIP-INs (roughly 32%) is immune to modulation through postsynaptic GABAbRs in both recording conditions.

In our current study, L2/3 VIP-INs were classified based on their electrophysiological features by implementing hierarchical clustering, as in recent years there has been a growing practice in cell-type classification using numerous clustering techniques for highly divergent neurons (Browne et al., 2020; Gouwens et al., 2019; Jiang et al., 2023; Scala et al., 2019 p.4; Yang et al., 2022).

Here we support the idea to categorize neurons in a quantitative manner (Hernáth et al., 2019; Komendantov et al., 2019) rather than qualitative, since there is a lack of consistency in terms of terminology of firing patterns across labs, creating confusion (Goldberg et al., 2004; Jiang et al., 2015, 2023; Kanigowski et al., 2023; Kawaguchi & Kubota, 1996; Ma et al., 2006; Otsuka & Kawaguchi, 2008; Prönneke et al., 2015). Clustering allows for unequivocal and unbiased characterization of the VIP-IN population. L2/3 VIP-INs show highly diverse morpho-electrophysiological and molecular properties (Jiang et al., 2023; Prönneke et al., 2015, 2019). Here, similarly to Jiang et al. (2023), we identified three electrophysiological types of L2/3 VIP-INs in the mouse barrel cortex. In general, type 1 has different properties from type 2 and 3. Types 2 and 3 VIP-INs display very similar basic membrane properties, which is also reflected in a cluster similarity on the dendrogram, where type 2 and type 3 VIP-INs emerge from the same main branch. Moreover, type 1 is sensitive to the modulation through GABAbRs both pre- and postsynaptically, whereas types 2 and 3 can be regulated only presynaptically. For the first time, our results uncover heterogeneity within the same molecularly distinctive interneuron population in terms of GABAbR modulation.

We started our experiments in classical conditions, using ACSF with a $Ca^{2+}$ concentration of 2.5 mM. In general, ACSF with 2–2.5 mM $Ca^{2+}$ has been used in most electrophysiological experiments in the field of neuroscience. Such a condition boosts all physiological processes that rely on $Ca^{2+}$ sensitivity, including neurotransmitter release (Borst, 2010; Feldmeyer et al., 2006; Finnerty et al., 1999; Ko et al., 2011; Lefort et al., 2009; Oláh et al., 2009; Pan et al., 2009; Scanziani, 2000; Silberberg & Markram, 2007). We found that in this condition, BK channels reduce the intrinsic excitability of all L2/3 VIP-IN types. We assume that the activation of GABAbRs suppresses BK channels probably indirectly through the inhibition of calcium channels (Cai et al., 2018; Garaycochea & Slaughter, 2016; Sun & Chiu, 1999) leading to an increase of intrinsic excitability of a subset of VIP-INs which were sensitive to the GABAbR agonist (type 1). Similar effects have been found in rat retinal ganglion cells, where GABAbRs suppress postsynaptic calcium channels leading to lower activity of BK channels, which in turn enhances the responses of these cells (Garaycochea & Slaughter, 2016). Also, GABAbRs have excitatory effects on the activity of calyces in rat semicircular canal cristae through suppression of BK channels (Ramakrishna & Sadeghi, 2020). Our study, for

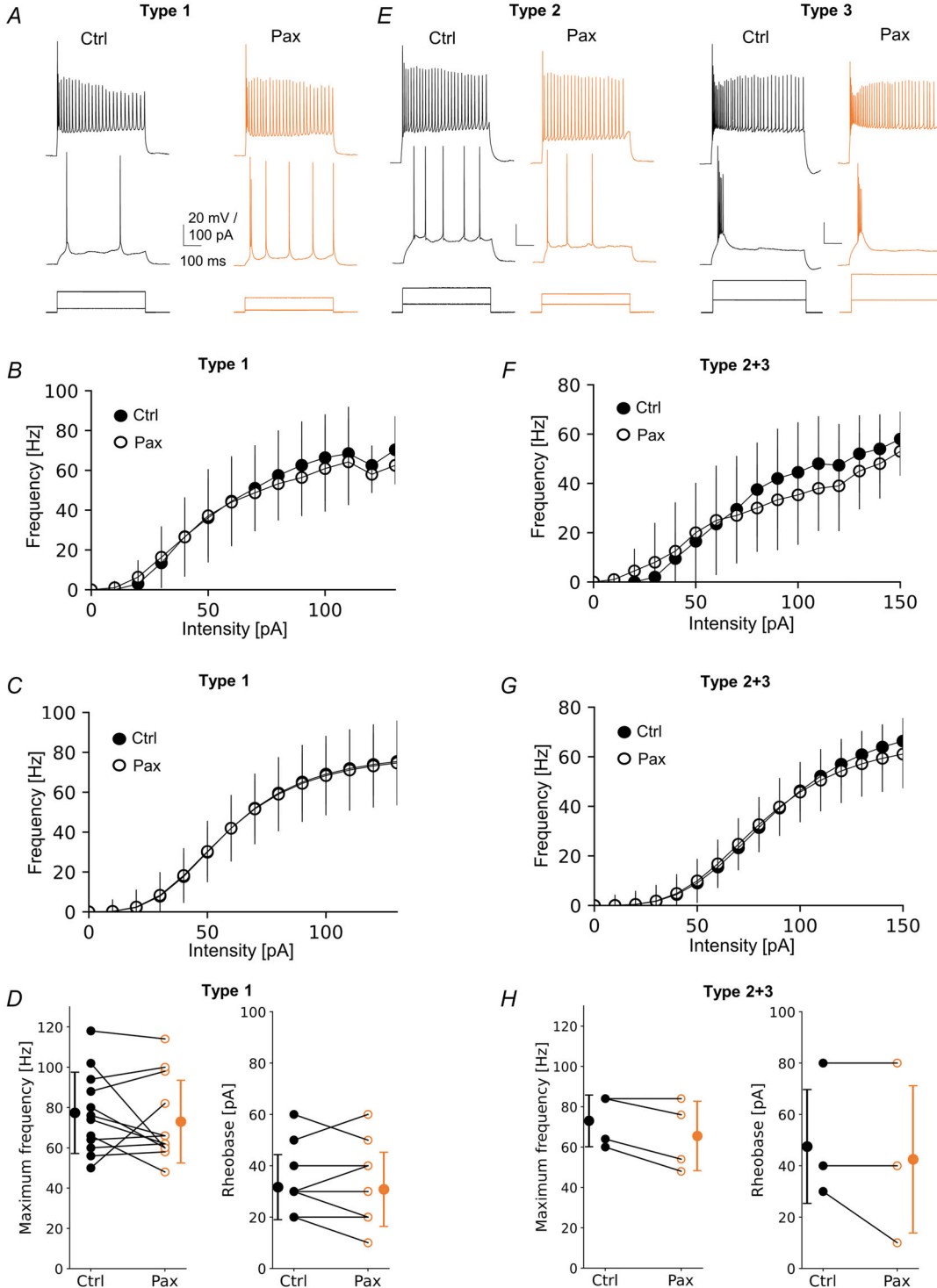

**Figure 8. BK channels do not regulate L2/3 VIP-IN intrinsic excitability at physiological Ca$^{2+}$ level**
*A*, examples of firing responses of type 1 upon a 500 ms-long pulse. Lower traces show step intensities of current injected to the soma, middle traces show neuronal firings at rheobase, upper traces represent maximum spike frequency in control ACSF (Ctrl, left) and after bath application of the BK channel antagonist (Pax, right). *B*, mean (±SD) firing frequency–current intensity curves for type 1 in control and after Pax ($P > 0.05$, paired *t* test, $n = 12$). *C*, Hill sigmoidal curves fitted to the data shown in *B*, $n = 12$. *D*, left, within-cell comparison and mean (±SD) maximum firing frequency of type 1 in control and after Pax ($P = 0.430$, paired *t* test, $n = 12$). Right, same but for rheobase ($P = 0.813$, Wilcoxon test, $n = 12$). *E–H*, same as *A–D* but for pooled raw data of types 2 and 3 VIP-INs ($P = 0.065$ paired *t* test and $P = 1.0$ Wilcoxon test, respectively, $n = 4$). [Colour figure can be viewed at wileyonlinelibrary.com]

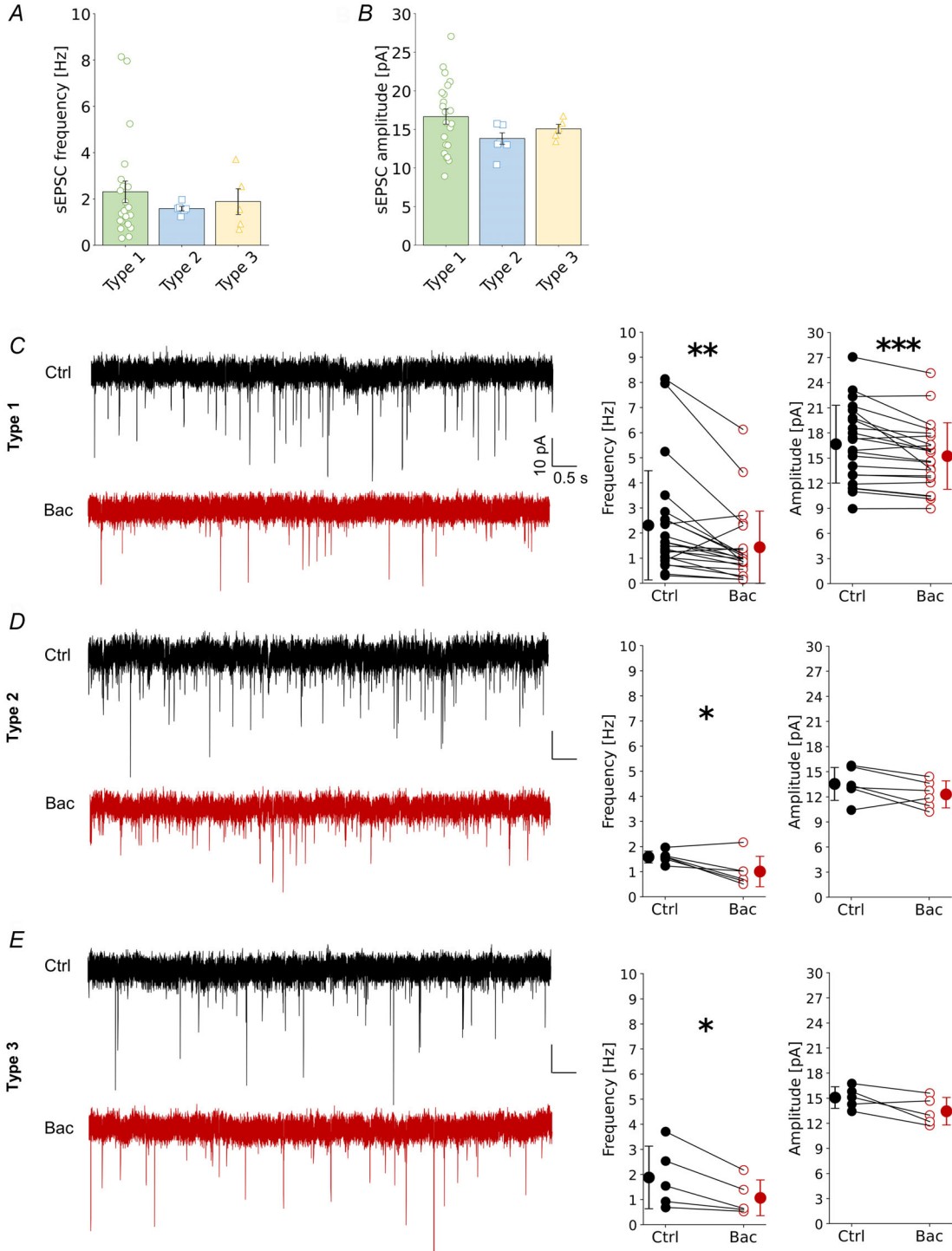

**Figure 9. GABAbRs suppress excitatory synaptic drive to L2/3 VIP-IN in an interneuron type-dependent manner when the Ca²⁺ level is high**

*A*, mean (±SD) sEPSC frequency and individual data points of all three types in control ACSF (*P* = 0.988, Kruskal–Wallis test, *n* = 22 for type 1, *n* = 6 for type 2 and *n* = 5 for type 3). *B*, same as *A* but for the amplitude (*P* = 0.240, Kruskal–Wallis test). *C*, left, example of spontaneous excitatory postsynaptic current (sEPSC) trace recorded in type 1 in control (Ctrl) and after baclofen (Bac). Middle, within-cell comparison and mean (±SD) sEPSC frequency of type 1 in control and after baclofen (*P* = 0.002, paired *t* test, *n* = 22). Right, same but for amplitude (*P* < 0.001, Wilcoxon test, *n* = 22). *D*, same as *C* but for type 2 (*P* = 0.033 and *P* = 0.107 paired *t* test, respectively, *n* = 6). *E*, Same as *C* but for type 3 (*P* = 0.033 and *P* = 0.065 paired *t* test, respectively, *n* = 5). **P* < 0.05, ***P* < 0.01, ****P* < 0.001. [Colour figure can be viewed at wileyonlinelibrary.com]

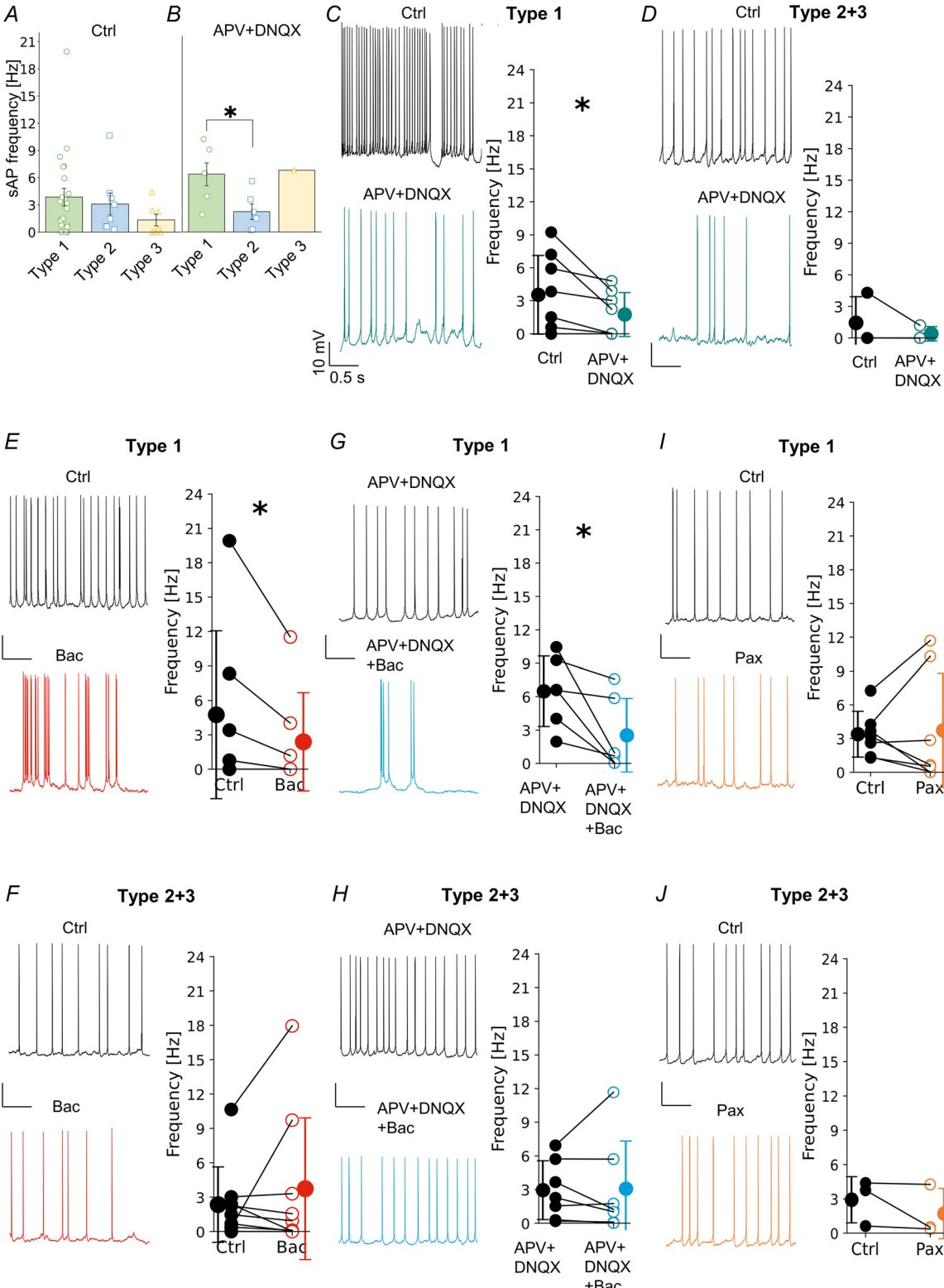

**Figure 10. L2/3 VIP-IN spontaneous firing in different recording conditions**

*A*, mean (±SD) spontaneous action potential (sAP) frequency and individual data points of all three types in control ACSF (*P* = 0.280, Kruskal–Wallis test, *n* = 23 for type 1, *n* = 9 for type 2 and *n* = 8 for type 3). *B*, same as *A* but in presence of the NMDAR and AMPAR antagonists, APV + DNQX (type 1 *vs.* type 2 *P* = 0.022, two sample *t* test, *n* = 6 for type 1, *n* = 6 for type 2 and *n* = 1 for type 3; change of statistical test due to low sample rate in type 3).

*C*, exemplar traces of spontaneous activity of type 1 in control (Ctrl, top) and after APV + DNQX (bottom). Within-cell comparison and mean (±SD) firing frequency of type 1 in these two conditions ($P = 0.031$, Wilcoxon test, $n = 8$). *D*, same as *C* but for type 2 + 3 ($P = 0.500$, Wilcoxon test, $n = 3$). *E*, same as *C* but in before (Ctrl) and after the GABAbR agonist (Bac) ($P = 0.043$, Wilcoxon test, $n = 7$). *F*, same as *E* but for type 2 + 3 ($P = 0.945$, Wilcoxon test, $n = 9$). *G*, same as *C* but in presence of the glutamate receptor antagonists (APV + DNQX) followed by ACSF with APV + DNQX + Bac ($P = 0.038$, paired *t* test, $n = 6$). *H*, same as *G* but for type 2 + 3 ($P = 0.886$, paired *t* test, $n = 7$). *I*, same as *C* but in control (Ctrl) and after the BK channel antagonist application (Pax) ($P = 0.810$, paired *t* test, $n = 7$). *J*, same as *I* but for type 2 + 3 ($P = 0.358$, paired *t* test, $n = 3$). *$P < 0.05$. [Colour figure can be viewed at wileyonlinelibrary.com]

the first time, shows excitatory effects of GABAbRs in the neocortex.

Our experiments also indicate a second mechanism by which GABAbRs have an enhancing effect on VIP-IN intrinsic excitability. Namely, presynaptic GABAbRs might inhibit GABA release and thus diminish the activity of postsynaptic GABAaRs on VIP-INs. L2/3 VIP-INs possess a relatively high intensity of synaptic and tonic GABAaR-mediated inhibition that reduces the intrinsic excitability of these interneurons (Bogaj et al., 2023). When presynaptic GABAbRs suppress GABA release from inhibitory interneurons targeting VIP-INs, this process might indirectly release VIP-INs from GABAaR-mediated inhibition. Summarizing, both mechanisms – through BK channels and GABAaRs – promote excitatory effects of GABAbRs on L2/3 VIP-INs in rACSF. The open question is under what circumstances these mechanisms are activated *in vivo*.

To shed light on mechanisms of GABAbR activation in more physiological conditions, we then performed *in vitro* recordings in mACSF. Literature data (Maffei et al., 2004; Sanchez-Vives & McCormick, 2000) and our previous experiments (Kanigowski et al., 2023; Urban-Ciecko et al., 2015, 2018) have shown that recordings in mACSF with physiological levels of extracellular $Ca^{2+}$ enable us to study neuronal function in the condition of the background slow oscillatory activity that mimics the quiet state *in vivo* (Maffei et al., 2004; Sanchez-Vives & McCormick, 2000). In this condition, L2/3 SST-INs (Fanselow et al., 2008; Kanigowski et al., 2023; Urban-Ciecko et al., 2015) and a subclass of L2/3 VIP-INs are highly active in providing GABA to the local network. In the context of network spontaneous activity, SST-IN firing suppresses excitatory synaptic transmission through presynaptic GABAbRs at connections between L2/3 pyramidal neurons (Urban-Ciecko et al., 2015) and between pyramidal neurons and SST-INs (Kanigowski et al., 2023).

Surprisingly, we found that in mACSF, GABAbRs have canonical inhibitory effects on the activity of L2/3 VIP-INs. Both intrinsic excitability and spontaneous firing of the main subgroup of VIP-INs (type 1) are reduced by these receptors. Again, type 2 and 3 were insensitive to GABAbR activation. We found that in mACSF, BK channels have no effect on either VIP-IN intrinsic excitability or their spontaneous firing.

Probably these channels are not activated effectively when extracellular $Ca^{2+}$ levels are too low.

Summarizing, effects of GABAbR activity on L2/3 VIP-INs depend on the recording conditions, presumably mostly on the $Ca^{2+}$ levels. Previously, we have found that intrinsic excitability of L2/3 SST- and PV-INs is differentially regulated through GABAbRs (Kanigowski et al., 2023). Namely, SST-IN excitability is resistant to GABAbRs, probably due to the fact that these receptors are not co-clustered with GIRK channels on SST-INs. Instead, GABAbRs are coupled with L-type channels affecting synaptic plasticity of excitatory inputs on SST-INs (Booker et al., 2018). In contrast, L2/3 PV-IN intrinsic excitability can be strongly suppressed by GABAbRs (Kanigowski et al., 2023).

Still, the switch between excitatory and inhibitory effects of GABAbRs on VIP-INs remains unclear. We hypothesize that BK channels determine VIP-IN activity by providing a switch mechanism between two modes of GABAbR action. In general, BK channels can have either an inhibitory (Hunsberger & Mynlieff, 2020; Ly et al., 2011; Matthews et al., 2008; Ramakrishna & Sadeghi, 2020) or an excitatory effect (Brenner et al., 2005; Gu et al., 2007; Ly et al., 2011; Shruti et al., 2008) on neuronal activity. When extracellular $Ca^{2+}$ levels are too low, BK channels are not activated on VIP-INs and activation of GIRK channels through postsynaptic GABAbRs leads to inhibitory effects on the activity of VIP-INs. However, when extracellular $Ca^{2+}$ levels increase, for example, as a consequence of a high network activity, now the activity of VIP-INs would paradoxically increase due to excitatory effects of GABAbRs. Because L2/3 VIP-INs inhibit other interneurons, mainly SST-INs but also PV-INs (Lee et al., 2013; Pfeffer et al., 2013; Pi et al., 2013), higher activity of VIP-INs would lead to stronger inhibition of other interneurons and in turn it would provide disinhibition of pyramidal neurons. On the contrary, our previous experiments (Kanigowski et al., 2023) have shown that intrinsic excitability of L2/3 SST-INs is almost completely immune to GABAbR modulation and very weakly sensitive to GABAaR-mediated inhibition; for this reason L2/3 SST-INs would not be a main target of GABA released by VIP-INs. In contrast to L2/3 SST-INs, PV-INs are strongly inhibited by GABAbRs (Kanigowski et al., 2023), suggesting that PV-INs might be strongly suppressed by activity of L2/3 VIP-INs. This process might

release pyramidal cells from somatic inhibition mediated by PV-INs and enhance the output signal from excitatory cells.

Our current and previous studies (Kanigowski et al., 2023; Urban-Ciecko et al., 2015) reveal specific mechanisms of GABAbR action on different interneuron populations in the neocortex. It is also worth mentioning that we found a subset of L2/3 VIP-INs that is not controlled postsynaptically by GABAbRs and thus these interneurons would provide effective inhibition to the network. It remains yet unknown what specific subclass is created by these VIP-INs that are insensitive to the regulation via GABAbRs.

As a potential limitation, our findings question the rationale for traditional patch-clamp recordings in standard ACSF. We may not exclude that the excitatory effect of GABAbRs in rACSF is merely an artifact. However, such an assumption would question hundreds of thousands electrophysiological experiments performed in standard ACSF. GABAbRs affect intrinsic excitability of L2/3 SST- and PV-INs in the same direction in regular and modified ACSF (Kanigowski et al., 2023), suggesting that our current results present a phenomenon that takes place *in vivo*.

In conclusion, we provide evidence for a new mechanism of GABAbR function in the neocortex. Taken together with previous data, we show that GABAbRs modulate neocortical circuits in a neuron- and synapse-specific manner. Diversity in GABAbR action may be considered as a crucial mechanism for shaping the balance between excitation and inhibition in neuronal networks. It has been suggested that heterogeneity of GABAergic interneurons allows for performing a broad range of complex computations in cortical circuits (Klausberger & Somogyi, 2008). Our results show a link between GABAbRs and BK channels. This phenomenon might be an essential mechanism in learning and plasticity processes, since BK channels act as a coincidence detector due to their $Ca^{2+}$ sensitivity.

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

## Additional information

### Data availability statement

The original contributions presented in the study are included in the article. The raw data will be openly available in OSF at https://osf.io/gw85z/.

## Competing interests

The authors declare they have no competing interests.

## Author contributions

K.B.: Methodology, formal analysis, investigation, visualization, writing – original draft preparation, review and editing. J.U.-C.: Conceptualization, methodology, validation, resources, data curation, supervision, project administration, funding acquisition, writing – original draft preparation and review and editing. All authors approved the final version of the manuscript. All authors agree to be accountable for all aspects of the work in ensuring that questions related to the accuracy or integrity of any part of the work are appropriately investigated and resolved. All persons designated as authors qualify for authorship, and all those who qualify for authorship are listed.

## Funding

This work was supported by the National Science Centre, Poland (2020/39/B/NZ4/01462 to J.U.-C).

## Keywords

barrel cortex, BK channels, excitability, GABAergic inhibition, somatosensory cortex, VIP interneurons

## Supporting information

Additional supporting information can be found online in the Supporting Information section at the end of the HTML view of the article. Supporting information files available:

**Peer Review History**

