## [Peer Review History · The Journal of Physiology]

Inhibition of BK channels by GABA_B receptors enhances intrinsic excitability of layer 2/3 VIP-interneurons in mouse neocortex

Karolina Bogaj and Joanna Urban-Ciecko

DOI: 10.1113/JP286439

Corresponding author(s): Joanna Urban-Ciecko (j.ciecko@nencki.edu.pl)

The following individual(s) involved in review of this submission have agreed to reveal their identity: Dirk Feldmeyer (Referee #2)

Review Timeline:

Submission Date:	16-Feb-2024
Editorial Decision:	11-Jun-2024
Revision Received:	22-Nov-2024
Accepted:	16-Jan-2025

Senior Editor: Katalin Toth

Reviewing Editor: Jean-Claude Béïque

Transaction Report:

Dear Dr Urban-Ciecko,

Re: JP-RP-2024-286439 "Inhibition of BK channels by GABA_B receptors enhances intrinsic excitability of layer 2/3 VIP-interneurons in mouse neocortex" by Karolina Bogaj and Joanna Urban-Ciecko

Thank you for submitting your manuscript to The Journal of Physiology. It has been assessed by a Reviewing Editor and by 2 expert referee and we are pleased to tell you that it is potentially acceptable for publication following satisfactory major revision.

REVISION CHECKLIST:

We look forward to receiving your revised submission.

Yours sincerely,

Katalin Toth
Senior Editor
The Journal of Physiology

REQUIRED ITEMS

- Author photo and profile. First or joint first authors are asked to provide a short biography (no more than 100 words for one author or 150 words in total for joint first authors) and a portrait photograph. These should be uploaded and clearly labelled together in a Word document with the revised version of the manuscript. See Information for Authors for further details.

- Your manuscript must include a complete Additional Information section, including competing interests; funding; author contributions and acknowledgements.

- The Journal of Physiology funds authors of provisionally accepted papers to use the premium BioRender site to create high resolution schematic figures. Follow this link and enter your details and the manuscript number to create and download figures. Upload these as the figure files for your revised submission. If you choose not to take up this offer, we require figures to be of similar quality and resolution. If you are opting out of this service to authors, state this in the Comments section on the Detailed Information page of the submission form. The link provided should only be used for the purposes of this submission. Authors will be charged for figures created on this premium BioRender account if they are not related to this manuscript submission.

- Please upload separate high-quality figure files via the submission form.

- Please ensure that any tables are editable and in Word format, and wherever possible, embedded in the article file itself.

- Please ensure that the Article File you upload is a Word file.

- Please include an Abstract Figure file, as well as the Figure Legend text within the main article file. The Abstract Figure is a piece of artwork designed to give readers an immediate understanding of the research and should summarise the main conclusions. If possible, the image should be easily 'readable' from left to right or top to bottom. It should show the physiological relevance of the manuscript so readers can assess the importance and content of its findings. Abstract Figures should not merely recapitulate other figures in the manuscript. Please try to keep the diagram as simple as possible and without superfluous information that may distract from the main conclusion(s). Abstract Figures must be provided by authors no later than the revised manuscript stage and should be uploaded as a separate file during online submission labelled as File Type 'Abstract Figure'. Please also ensure that you include the figure legend in the main article file. All Abstract Figures should be created using BioRender. Authors should use The Journal's premium BioRender account to export high-resolution images. Details on how to use and access the premium account are included as part of this email.

- Please include a full title page as part of your main article (Word) file, which should contain the following: title, authors,

affiliations, corresponding author name and contact details, keywords, and running title.

- Please ensure that all figures and tables have a title and legend, and that they have been cited within the main article text.

EDITOR COMMENTS

Reviewing Editor:

This manuscript has been assessed by two expert reviewers. In general, they both provide a positive assessment of this study, both in terms of general interest and of overall quality of the data. They however raised several (and at times converging) issues that need to be satisfactorily addressed before publication. While several issues can be appropriately dealt with with additional analysis and re-writing, some may require additional experiments. The authors should note that J Physiol does not allow inclusion of supplemental figures, so all data required to be illustrated in figure format should be done as part of the main figures.

Please also see 'Required Items' above.

REFEREE COMMENTS

Referee #1:

This is a review of the manuscript entitled "Inhibition of BK channels by GABA_B receptors enhances intrinsic excitability of layer 2/3 VIP-interneurons in mouse neocortex" by Bodaj and Urban-Ciecko. In this manuscript they perform whole cell current- and voltage- clamp recordings in layer L2/3 neurons of the mouse primary somatosensory cortex. They tested the hypothesis asking if GABA_BRs can modulate the activity of vasoactive intestinal polypeptide-expressing interneurons (VIP-INS). Using principal component analysis and K-Means clustering they conclude that 2 out of 3 distinct population of VIP-INS were indeed sensitive to GABA_BR activation. They then characterize this phenomenon by testing high and low extracellular Ca⁺⁺ concentrations. From this they conclude that in elevated extracellular Ca⁺⁺, GABA_BR activation enhances intrinsic excitability through an indirect inhibition of BK channels and a reduction in GABA_BR inhibition. While in the low Ca⁺⁺ case, they saw a more classic inhibitory effect of GABA_BR activation.

For me, the key contribution here is they have further expanded the role of metabotropic GABA receptors in modulation the excitability of VIP-Ins. In addition their examination of high and low Ca⁺⁺ does improve on the realization that many in vitro results may be dependent on chosen (potentially non-physiological) experimental configurations.

I do have some major and minor concern I would like to see addressed.

Major Concerns

The results begin with references to supplemental figure 1. In this figure legend, the number of recordings and mice are reported. Please move this information into the main results section. For example, "n=57 cells in 42 mice". Was this the total number of neurons used to cluster their 3 types?

Because all the analysis relies on the initial clustering into 3 cell types, I would vote to have all or part of supplemental figure 1 moved into the manuscript as figure 1.

I do have concerns over the limited choice of features to fit to generate a dendrogram, then PCA, and k-mean clustering

seem limited and rather arbitrary. Because this analysis results in the clustering of cells into three main type that are used throughout the manuscript, what if additional parameters were added? Would that effect the clustering, the number of groups, and the categorization of a particular cell(recording) into a group?

For example, the authors could try and include other physiological relevant features such as: (i) AP half width, (ii) threshold potential, (iii) AP peak, (iv) AP fast AHP, (v) AP slow AHP, (vi) first second AP frequency ratio, (vii) first last AP frequency ratio, etc. etc. Another feature would be burst firing versus adaptive spikes (see supp s1d). Another feature they could add to the clustering algorithm is information they already extract from the F-I curves such as steepness and midpoint.

I ask the authors include some number of these and rerun their clustering to ensure the reader that their three types they then analyze throughout the remaining manuscript are representative of actual phenotypic type. Does adding additional electrophysiology parameter result in similar or completely different clustering?

Fig s2a. There is a significant difference in resting membrane potential between cluster types 1-2, 1-3, and 2-3. Yet, in the methods they describe how they held membrane potential constant with DC injections (am I correct). Although you will get a clustering based on a parameter like resting Vm regardless of how it was controlled or not. Is this meaningful to include resting Vm in the clustering if it was experimentally controlled?

"Input resistance was the most important factor for cluster distribution". I would guess that if they included other relevant features in their clustering that this might not be the case. I would like to see some exploration of additional parameters that go into the clustering (see above).

"SAG amplitude" is used throughout but is never described why it might be important. It is calculated from the trajectory of the membrane potential in response to a hyperpolarizing pulse. Is this supposed to be a measure a current such as I_H? Please add an explanation of the biophysical property that this "SAG" analysis is measuring.

In figure S1d they show examples of Type I/II/II. Then in figure 1 they show examples of type I/II/II. I really can't quickly/rapidly/intuitively see what firing phenotype I am looking for in types I/II/II? The examples look different. This relates to me concern about the limited set of parameters that were used to classify these neurons into different firing phenotype (via dendrogram, pca, and k-means),

Minor Concerns

I would like the authors to make a better distinction on high and low Ca⁺⁺, which one is considered physiological? For example, in the introduction, "In the condition of elevated extracellular Ca₂₊ level" and "in the condition of physiological low Ca₂₊ level".

Can the authors provide the breakdown of results for male and female mice? Is there any hint there is a difference in the results between sex? The principal component analysis and k-mean clustering is perfectly conducive to this as sex can be added as another variable to fit. By another variable, I mean like in "methods - clustering", they have features like rheobase level and membrane capacitance. Animal sex would be another feature. Although not needed for this manuscript. I would strongly suggest the authors start including male and female sex as a potentially important parameter.

Not needed for this manuscript but I want to stress to the authors they should also include morphological parameters in their clustering algorithms. This is common practice (Ascoli, 2018) and they already have fluorescent labelled neurons they could image and quantify after fixing the tissue.

They report that recordings were performed with "warm" ACSF at 31-32 deg Celsius. Can they provide information on how ACSF was warmed? Was it a feedback in-line temperature controller or other?

Can they better explain "Test pulse (100 ms square pulse, -10 pA) was added within the protocol as to measure input resistance (R_{in})". Incorporate into the text and explain better?

Methods

1) The "Data Analysis" section of the methods needs to be rewritten/converted to paragraphs. It is currently more like a bulleted list.

2) How were action potentials detected, all is said is "Spontaneous action potential firings (sAPs) were measured". Was this done in pClamp? Was it using a voltage or dv/dt threshold, was there a refractory period, was there a minimal or maximum AP amplitude requirement? Although rather trivial, we need to know how APs were detected.

3) Same for sEPSC, there is no mention on how spontaneous EPSCs were detected. Please include a description of how this was done.

4) They mention Python was used, but how spikes and EPSCs were detected and their parameters extracted and analyzed is important.

"F-I curves were compared step-by-step using raw data". Can the authors further clarify describe this. What is "Step-by-step"? Is it for each current step? Otherwise, might not be important and could just remove the sentence.

"Proper number of clusters was assessed consistently with the dendrogram and additionally evaluated with elbow method". Can the authors elaborate on the "elbow" method with 1-2 additional descriptive sentences.

Likewise, "Factor vector map was introduced to the PCA in order to obtain variables which are crucial for each cluster assignment". Can the authors elaborate on what a "factor vector map" is?

"Probes were initially tested for normal distribution". What is meant by "Probes"?

Figure S1

S1C, the individual cells are labeled with symbols, can you indicate the clustered "type" with unique symbols. They are all currently and "x" symbol

S1D, likewise, in your pie chart, can you indicate which portions are which type? There is no legend for the 'hash' patterns in each portion of the pie chart.

Referee #2:

This is an interesting study on the effect of GABA_B receptor (GABA_BR) modulation of large conductance Ca²⁺-activated K⁺ channels (BK channels) in VIP interneurons in the primary somatosensory barrel field. The authors used a transgenic animal in which VIP interneurons are specifically labelled. They identified three distinct VIP cell types based on their sensitivity to GABA_BR modulation and BK channel activation. The AP firing frequency of type 1 and type 2 VIP cells is modulated by GABA_BRs, whereas that of type 3 neurons is not. In the two GABA_BR-sensitive groups, the effect of GABA is mediated by an indirect inhibition of BK channels, as indicated by the experiments with paxillin. The experiments showed that Ca²⁺ influences the direction of the GABA_BR effect, increasing excitability at high Ca²⁺ and decreasing excitability and firing frequency at low Ca²⁺. Unfortunately, despite the novel findings, the study is somewhat incoherent and, in my opinion, would greatly benefit from additional experiments that address the point of VIP cell type classification.

Specific points

The grammar and style is often inaccurate and should be checked thoroughly. There is an overuse of the definite article and some words are ill-defined.

There is a lack of coherence in presenting the experimental design and data. The authors specifically mention that they were able to identify three different VIP cell types but show this important data only in suppl. figures S1 and S2. I suggest to combine the two figures into one and move the resulting figure to the main manuscript.

The comparison of the GABA_BR-mediated effect is confusing and makes the manuscript difficult to read because from figure to figure different VIP cell types are compared. For example, in figure 1, the existence of three VIP cell types is described briefly, in figure 2 only type 2 and type 3 are compared, in figure 3, the comparison is between type 1 and type 3 and figure 4 for again between type 2 and type 3. This is rather disconcerting and shows a lack of focus. In this context, colour-coding of recordings from the different VIP cell types may help a reader to identify what has been compared. Finally the authors often pool data from type 1 and type 2 VIP cells without mentioning via this can be done. This requires a justification.

In the light of the extensive morphological and transcriptomic analysis of VIP cell types it would have been helpful to match the type 1 to 3 VIP cell types. I believe it would greatly enhance the impact of the manuscript if one could show a link between the neuromodulatory and morphological cell type. For example, are the GABA_BR-sensitive neurons VIP/CCK neurons and the unresponsive ones VIP/ChaT neurons? These experiments just require biocytin filling during recording and e.g. ChaT and CCK immunohistochemistry.

The patch clamp recordings from appear to be not properly compensated for the series resistance. and some action potentials (MPs) are of unusually small amplitude. Please provide the series and/or access resistance for all recordings, and the cut-off for analysis.

In the figure, many of the comparison histograms show no significant differences. I suggest to relegate at least some of those to the supplemental material.

The comparison of 'regular' ACSF (2.5 mM Ca²⁺) with low Ca²⁺ ACSF is probably exaggerating the effect described. Firstly, 2.5 mM Ca²⁺ is higher than the commonly used ACSF (2 mM Ca²⁺) and too low in the 1 mM Ca²⁺ concentration in the 'Low Ca²⁺' ACSF. In fact, 1 mM Ca²⁺ will result in ~0.8 mM free Ca²⁺. Have the authors looked at direct effects (such as a depolarisation). However, in experiments on human tissue realistic values for the free extracellular Ca²⁺ concentration in the CSF have been measured with ion-sensitive microelectrodes to be between 1.1 to 1.3 mM, i.e. 1.3 to 1.5 total Ca²⁺. It would be of interest to now whether the GABA_BR effect is excitatory or inhibitory under this condition. Could the authors please comment!

Specific points

Page 7, 'Curve fitting'

Which data were used for curve fitting?

Page 7, last para.

Please quote the original papers for cluster analysis, not only a review paper.

Page 8, last para.

Please quote the original paper when you introduce 'Ward's method'.

Page 14, 2nd paragraph

A good way to identify whether the GABA_BR effects were mediated via pre- and/or postsynaptic receptors. Has this been tried? If yes, this should be mentioned here.

Page 17, last two paragraphs

It appears that the effect of GABA_BR if pre- and or has indeed been analysed but no figure is shown. The relevant figure is in the Supplemental Materials. It should be moved to the main text body,

END OF COMMENTS

Ms. JP-RP-2024-286439

Inhibition of BK channels by GABA_B receptors enhances intrinsic excitability of layer 2/3 VIP-interneurons in mouse neocortex

by K. Bogaj and J. Urban-Ciecko

This is an interesting study on the effect of GABA_B receptor (GABA_BR) modulation of large conductance Ca²⁺-activated K⁺ channels (BK channels) in VIP interneurons in the primary somatosensory barrel field. The authors used a transgenic animal in which VIP interneurons are specifically labelled. They identified three distinct VIP cell types based on their sensitivity to GABA_BR modulation and BK channel activation. The AP firing frequency of type 1 and type 2 VIP cells is modulated by GABA_BRs, whereas that of type 3 neurons is not. In the two GABA_BR-sensitive groups, the effect of GABA is mediated by an indirect inhibition of BK channels, as indicated by the experiments with paxillin. The experiments showed that Ca²⁺ influences the direction of the GABA_BR effect, increasing excitability at high Ca²⁺ and decreasing excitability and firing frequency at low Ca²⁺. Unfortunately, despite the novel findings, the study is somewhat incoherent and, in my opinion, would greatly benefit from additional experiments that address the point of VIP cell type classification.

Specific points

The grammar and style is often inaccurate and should be checked thoroughly. There is an overuse of the definite article and some words are ill-defined.

There is a lack of coherence in presenting the experimental design and data. The authors specifically mention that they were able to identify three different VIP cell types but show this important data only in suppl. figures S1 and S2. I suggest to combine the two figures into one and move the resulting figure to the main manuscript.

The comparison of the GABA_BR-mediated effect is confusing and makes the manuscript difficult to read because from figure to figure different VIP cell types are compared. For example, in figure 1, the existence of three VIP cell types is described briefly, in figure 2 only type 2 and type 3 are compared, in figure 3, the comparison is between type 1 and type 3 and figure for again between type 2 and type 3 in figure 4. This is rather disconcerting and shows a lack of focus. In this context, colour-coding of recordings from the different VIP cell types may help a reader to identify what has been compared. Finally the authors often pool data from type 1 and type 2 VIP cells without mentioning via this can be done. This requires a justification.

In the light of the extensive morphological and transcriptomic analysis of VIP cell types it would have been helpful to match the type 1 to 3 VIP cell types. I believe it would greatly enhance the impact of the manuscript if one could show a link between the neuromodulatory and morphological

cell type. For example, are the GABA_BR-sensitive neurons VIP/CCK neurons and the unresponsive ones VIP/ChaT neurons? These experiments just require biocytin filling during recording and staining and perhaps but not necessarily ChaT and CCK immunohistochemistry.

It appear in the original patch clamp recordings that there was no proper series resistance compensation. In addition, and some action potentials (MPs) have unusually small amplitudes. Please provide the series and/or access resistance for all recordings, and under which condition data were excluded from the analysis.

In the figure, many of the comparison histograms show no significant differences. I suggest to relegate at least some of those to the supplemental material.

The comparison of 'regular' ACSF (2.5 mM Ca²⁺) with low Ca²⁺ ACSF is probably exaggerating the effect described. Firstly, 2.5 mM Ca²⁺ is higher than the commonly used ACSF (2 mM Ca²⁺) and too low in the 1 mM Ca²⁺ concentration in the 'Low Ca²⁺' ACSF. In fact, 1 mM Ca²⁺ will result in ~0.8 mM free Ca²⁺. Have the authors looked at direct effects (such as a depolarisation). However, in experiments on human tissue realistic values for the free extracellular Ca²⁺ concentration in the CSF have been measured with ion-sensitive microelectrodes to be between 1.1 to 1.3 mM, i.e 1.3 to 1.5 total Ca²⁺. It would be of interest to now whether the GABA_BR effect is excitatory or inhibitory under this condition. Could the authors please comment!

Specific points

Page 7, 'Curve fitting'

Which data were used for curve fitting?

Page 7, last para.

Please quote the original papers for cluster analysis, not only a review paper.

Page 8, last para.

Please quote the original paper when you introduce 'Ward's method'.

Page 14, 2nd paragraph

I good way to identify whether the GABA_BR effects were mediated via pre- and/or postsynaptic receptors. Has this been tried? If yes, this should be mentioned here.

Page 17, last two paragraphs

It appears that the effect of GABA_BR if pre- and or has indeed been analysed but no figure is shown. The relevant figure is in the Supplemental Materials. It should be moved to the main text body,

Dear Editor and Reviewers,

We thank the reviewers for their thoughtful and constructive suggestions.

Reviewers pointed out that this study is interesting, presenting valuable new information that advances our understanding of the cortical GABAergic system. In response to reviewers' suggestions, we have now performed new analysis and experiments. We have clarified some methodological details and prepared 10 main figures because J. Physiology does not allow inclusion of supplementary figures.

Altogether, we are grateful for the comments provided and we think that this report is substantially improved. Attached please find our response to specific comments and the revised manuscript. We hope that you will find this work suitable for publication in Journal of Physiology. Please do not hesitate to contact me if you have any questions or concerns.

Sincerely,

Joanna Urban-Ciecko

EDITOR COMMENTS

Reviewing Editor:

This manuscript has been assessed by two expert reviewers. In general, they both provide a positive assessment of this study, both in terms of general interest and of overall quality of the data. They however raised several (and at times converging) issues that need to be satisfactorily addressed before publication. While several issues can be appropriately dealt with additional analysis and re-writing, some may require additional experiments. The authors should note that J Physiol does not allow inclusion of supplemental figures, so all data required to be illustrated in figure format should be done as part of the main figures.

Please also see 'Required Items' above.

Point-by-point responses:

Referee #1:

1. The results begin with references to supplemental figure 1. In this figure legend, the number of recordings and mice are reported. Please move this information into the main results section. For example, "n=57 cells in 42 mice". Was this the total number of neurons used to cluster their 3 types?

In this study, the total number of cells for clustering was 176. However, neurons in each experiment with different drug administration and different conditions (regular ACSF or modified ACSF) were clustered separately. Different composition of ACSF (modified vs regular) affect the membrane properties. Furthermore, our investigation was performed over the course of 3 years, which also impacts data variation. In this case, for Figure 1 (supplementary Figure S1 in the previous version), the number of neurons was 50 in 37 mice. The discrepancy in numbers between old and new versions comes from the fact that now we performed new clustering based

on 18 electrophysiological features, as suggested. We discarded 7 cells from 5 mice because we were unable to collect all the parameters from these cells for new clustering.

2. Because all the analysis relies on the initial clustering into 3 cell types, I would vote to have all or part of supplemental figure 1 moved into the manuscript as figure 1.

We moved the supplementary Figure S1 to main figures. This is now Figure 1. Due to journal's requirements, there is no supplement file at all. All the data were moved to main figures or to the main text.

3. I do have concerns over the limited choice of features to fit to generate a dendrogram, then PCA, and k-mean clustering seem limited and rather arbitrary. Because this analysis results in the clustering of cells into three main type that are used throughout the manuscript, what if additional parameters were added? Would that effect the clustering, the number of groups, and the categorization of a particular cell(recording) into a group? For example, the authors could try and include other physiological relevant features such as: (i) AP half width, (ii) threshold potential, (iii) AP peak, (iv) AP fast AHP, (v) AP slow AHP, (vi) first second AP frequency ratio, (vii) first last AP frequency ratio, etc. etc. Another feature would be burst firing versus adaptive spikes (see supp s1d). Another feature they could add to the clustering algorithm is information they already extract from the F-I curves such as steepness and midpoint.

I ask the authors include some number of these and rerun their clustering to ensure the reader that their three types they then analyze throughout the remaining manuscript are representative of actual phenotypic type. Does adding additional electrophysiology parameter result in similar or completely different clustering?

We agree that the initial number of parameters used for clustering was rather limited (only 6). As suggested, now the clustering is performed on 18 parameters as follows: rheobase, membrane capacitance, resting membrane potential, input resistance, maximum spike frequency, sag amplitude, presence of rebound spike, burst or adaptive spiking, mean firing frequency, medium and fast afterhyperpolarization (AHP), midpoint and steepness of a curve, membrane time constant, threshold, spike amplitude change, half-width of action potential (AP), mean interevent interval (ISI).

The result of new clustering is generally similar with the previous one. Still, there are three main clusters (see Figures 1 and 2). Of course, some cells shifted between clusters but generally the conclusion is similar. The percentage of cells that are sensitive to postsynaptic modulation via GABA_BRs is similar - 68% in new clustering (Type 1) vs. 70% in the previous clustering (Types 1 and 2).

4. Fig s2a. There is a significant difference in resting membrane potential between cluster types 1-2, 1-3, and 2-3. Yet, in the methods they describe how they held membrane potential constant with DC injections (am I correct). Although you will get a clustering based on a parameter like resting V_m regardless of how it was controlled or not. Is this meaningful to include resting V_m in the clustering if it was experimentally controlled?

For intrinsic excitability experiment, membrane potential was indeed controlled, held at -70mV. However we believe that including resting membrane potential in clustering is valid, since it is a passive membrane property just like input resistance or membrane time constant. V_m is 11th feature that shaped cluster assignment (see Figure 1B). Moreover, the inclusion of V_m in clustering has been previously reported, even if then the cells were recorded at V_{hold} (Nassar *et al.*, 2024).

5. "Input resistance was the most important factor for cluster distribution". I would guess that if they included other relevant features in their clustering that this might not be the case. I would like to see some exploration of additional parameters that go into the clustering (see above).

As suggested, we added new parameters to the clustering and now the most important feature for clustering was membrane time constant (Tau). Rin is the 5th most relevant property for cluster distribution (see Figure 1B).

6. "SAG amplitude" is used throughout but is never described why it might be important. It is calculated from the trajectory of the membrane potential in response to a hyperpolarizing pulse. Is this supposed to be a measure a current such as I_H? Please add an explanation of the biophysical property that this "SAG" analysis is measuring.

Yes, sag amplitude is indeed a measure of I_h currents, we added his information to the text and explained how it was calculated. Now the sentence reads:

"Sag amplitude, which is a measure of I_h current, was calculated as a difference between minimum and steady-state value of the voltage in the hyperpolarizing square pulse (-200 pA, 500 ms)." (Methods, page 7)

7. In figure S1d they show examples of Type I/II/III. Then in figure 1 they show examples of type I/II/III. I really can't quickly/rapidly/intuitively see what firing phenotype I am looking for in types I/II/III? The examples look different. This relates to me concern about the limited set of parameters that were used to classify these neurons into different firing phenotype (via dendrogram, pca, and k-means),

Figure S1D (now Figure 1D) shows examples of most frequent firing phenotypes in each cluster. In general, there is no consistency, that each cluster has specific/characteristic firing pattern. The point of clustering was to show that comparing neurons based on their electrophysiological properties is more relevant than grouping cells based on their firing phenotypes, which can be subjective. Figures 3-8 show exemplary traces before and after drug administration to better visualize how the drug changes firing properties. For the consistency, now we show examples of firings for all the 3 Types across all the experiments.

Minor Concerns

8. I would like the authors to make a better distinction on high and low Ca⁺⁺, which one is considered physiological? For example, in the introduction, "In the condition of elevated extracellular Ca²⁺ level" and "in the condition of physiological low Ca²⁺ level".

More physiological solution is ACSF with low Ca²⁺ concentration (1mM), called in literature as modified ACSF (Sanchez-Vives & McCormick, 2000; Maffei *et al.*, 2004). We unified our nomenclature for Ca²⁺ concentrations across the whole text. Now it reads: high Ca²⁺ levels (in regards to 2.5mM) and physiological Ca²⁺ (in regards to 1mM). Also, we added this explanation to key points.

9. Can the authors provide the breakdown of results for male and female mice? Is there any hint there is a difference in the results between sex? The principal component analysis and k-mean clustering is perfectly conducive to this as sex can be added as another variable to fit. By another variable, I mean like in "methods - clustering", they have features like rheobase level and membrane capacitance. Animal sex would be another feature. Although not needed for this manuscript. I would strongly suggest the authors start including male and female sex as a

potentially important parameter.

We agree that this is an interesting aspect to consider in the future. We performed experiments on both sexes. However, when we added this parameter to clustering, the dendrogram did not change, cluster assignment remained the same as before. Here we show feature importance score, accuracy 80% (sex is located at the bottom of the list):

1. Tau	0.248
2. Midpoint	0.120
3. Threshold	0.084
4. Rheobase	0.067
5. Amplitude_change	0.063
6. Rin	0.061
7. Sag	0.056
8. Cp	0.056
9. AHP_fast	0.046
10. Steepness	0.041
11. Mean_ISI	0.034
12. Mean_frequency	0.027
13. AHP_slow	0.021
14. Vm	0.021
15. burst/adapt	0.020
16. Max_frequency	0.014
17. AP_half-width	0.013
18. Sex	0.004
19. Rebound_spike	0.003

10. Not needed for this manuscript but I want to stress to the authors they should also include morphological parameters in their clustering algorithms. This is common practice (Ascoli, 2018) and they already have fluorescent labelled neurons they could image and quantify after fixing the tissue.

We agree that morphological reconstructions would be worth performing. We can hypothesize that GABA_BR sensitivity might be correlated with the morphology of VIP-INs. However, for now, we think that this is a separate story for the next project.

11. They report that recordings were performed with "warm" ACSF at 31-32 deg Celsius. Can they provide information on how ACSF was warmed? Was it a feedback in-line temperature controller or other?

The recovery chamber was heated with a custom-made heating plate. Recording ACSF was warmed up by using an in-line heater with temperature control system (Harvard Apparatus, USA). We incorporated this information to the text.

11. Can they better explain "Test pulse (100 ms square pulse, -10 pA) was added within the protocol as to measure input resistance (Rin)". Incorporate into the text and explain better?

Now the text reads:

“To measure input resistance (R_{in}), a test pulse (-10 pA lasting 100 ms) was applied at the beginning of every sweep in the current-clamp mode.” (Methods, page 6)

Methods

12. The "Data Analysis" section of the methods needs to be rewritten/converted to paragraphs. It is currently more like a bulleted list.

We rewrote this part. We hope now the text has been improved.

13. How were action potentials detected, all is said is "Spontaneous action potential firings (sAPs) were measured". Was this done in pClamp? Was it using a voltage or dv/dt threshold, was there a refractory period, was there a minimal or maximum AP amplitude requirement? Although rather trivial, we need to know how APs were detected. Same for sEPSC, there is no mention on how spontaneous EPSCs were detected. Please include a description of how this was done. They mention Python was used, but how spikes and EPSCs were detected and their parameters extracted and analyzed is important.

All the analyses of electrophysiological parameters were carried out using standard tools in Clampfit software (Molecular Devices, USA). sAPs were detected with threshold-based search tool, with the criterion of spike overshooting 0 mV threshold. sEPSCs were analyzed with template search tool, in which templates were manually fitted to identify events, with cutoff at 5 pA amplitude. Python was used for clustering, curve fitting, statistical analysis and visualization, not for event detection.

14. "F-I curves were compared step-by-step using raw data". Can the authors further clarify describe this. What is "Step-by-step"? Is it for each current step? Otherwise, might not be important and could just remove the sentence.

Yes, the comparison was done for each current step before and after drug applications. We removed this sentence as suggested.

15. "Proper number of clusters was assessed consistently with the dendrogram and additionally evaluated with elbow method". Can the authors elaborate on the "elbow" method with 1-2 additional descriptive sentences.

We no longer use elbow method and this sentence was removed. We changed approach – types are based on hierarchical clustering, so the dendrogram provides information about the clusters assignment. We decided to use dendrogram for clustering as it is an unsupervised method, and it's more reliable than k-means clustering with such limited dataset. Hierarchical clustering is more stable, whereas k-means can produce different results depending on centroids (Xu & WunschII, 2005). Hierarchical clustering is also a common approach in literature (Muñoz-Manchado *et al.*, 2018; Jiang *et al.*, 2023; Nassar *et al.*, 2024).

16. Likewise, "Factor vector map was introduced to the PCA in order to obtain variables which are crucial for each cluster assignment". Can the authors elaborate on what a "factor vector map" is?

We added the explanation to the Method section as follows:

“Then we plotted a vector factor map showing relations between all electrophysiological properties in the PCA matrix. Vector length and direction represent feature correlation with principal component. Vectors with opposite directions are negatively correlated to each other, whereas the closer the vectors are to each other, the related properties are more correlated (Jolliffe & Cadima, 2016).” (Methods, pages 8 and 9)

17. "Probes were initially tested for normal distribution". What is meant by "Probes"?

Corrected to “data”:

“Data were initially tested for normal distribution with Shapiro-Wilk test.”

18. Figure S1, S1C, the individual cells are labeled with symbols, can you indicate the clustered "type" with unique symbols. They are all currently and "x" symbol S1D, likewise, in your pie chart, can you indicate which portions are which type? There is no legend for the 'hash' patterns in each portion of the pie chart.

In Figure S1C (now Figure 1C), types are marked as circle 'o' (Type 1), square '□' (Type 2) and triangle 'Δ' (Type 3). We no longer use 'x' which marked cluster centroids, since we don't show k-means (see explanation in question 15). Furthermore, we color-coded types – Type 1 is green, Type 2 is blue and Type 3 is yellow. In pie chart (Figure 1D), we also colored example traces and pie sections so it is easier to follow. Additionally, in figures showing before and after drug application, we colored traces and dots corresponding to the drug administration. We hope it allows readers for better understanding the graphs.

Referee #2:

1. The grammar and style is often inaccurate and should be checked thoroughly. There is an overuse of the definite article and some words are ill-defined.

Corrected.

2. There is a lack of coherence in presenting the experimental design and data. The authors specifically mention that they were able to identify three different VIP cell types but show this important data only in suppl. figures S1 and S2. I suggest to combine the two figures into one and move the resulting figure to the main manuscript.

As suggested also by the Referee #1, we moved first two supplementary figures to main figures. Now clustering results are presented as Figure 1 and Figure 2.

3. The comparison of the GAB_BR-mediated effect is confusing and makes the manuscript difficult to read because from figure to figure different VIP cell types are compared. For example, in figure 1, the existence of three VIP cell types is described briefly, in figure 2 only type 2 and type 3 are compared, in figure 3, the comparison is between type 1 and type 3 and figure for again between type 2 and type 3 in figure 4. This is rather disconcerting and shows a lack of focus. In this context, colour-coding of recordings from the different VIP cell types may help a reader to identify what has been compared. Finally the authors often pool data from type 1 and

type 2 VIP cells without mentioning via this can be done. This requires a justification.

We apologize for the confusion. Now we present example traces for all types in each figure. Following Referee's #1 comments, we analyzed more electrophysiological parameters and implemented them to new clustering (18 in total), and decided to cluster dataset using Ward's method hierarchical clustering. That approach creates stable cluster divisions, unlike k-means clustering (Xu & WunschII, 2005). Thus now the clusters shuffled, but the conclusion remains - around 68% of the cells are sensitive to GABA_BR modulation pre- and postsynaptically, whereas the rest can be regulated via these receptors only presynaptically. The 68% subset is now Type 1, 12% Type 2 and 20% Type 3 (See Figure 1 A,C). Now we pool Types 2 and 3, since they respond to drug administration in similar ways and they emerge from the same dendrogram branch. In the text we write:

"However, here and for the following experiments, we pooled Types 2 and 3 into one cluster, because these types belong to the same dendrogram clade (Fig. 1A) and responded similarly to GABA_BR activation with baclofen (Fig. 3E-L)." (Results, page 12)

We also used color-coding for on-line color figures. And add more figure legends for black-and-white figures to clarify data presentation.

4. In the light of the extensive morphological and transcriptomic analysis of VIP cell types it would have been helpful to match the type 1 to 3 VIP cell types. I believe it would greatly enhance the impact of the manuscript if one could show a link between the neuromodulatory and morphological cell type. For example, are the GABA_BR-sensitive neurons VIP/CCK neurons and the unresponsive ones VIP/ChaT neurons? These experiments just require biocytin filling during recording and e.g. ChaT and CCK immunohistochemistry.

We totally agree that our findings would definitely benefit from such analyses. It could broaden our understanding of mechanism underlying VIP-IN sensitivity to GABA_BRs. Indeed, Type 1 could express CCK (Hioki *et al.*, 2018) or CR (Kawaguchi, 1997), whereas Types 2 and 3 could express ChaT (Dudai *et al.*, 2020). We admit that morphological analysis would boost the impact of our paper. Unfortunately it requires to redo all of the experiments. Thus, definitely morphological and transcriptomic analyses as well as immunohistochemistry are worth doing in the future.

5. The patch clamp recordings from appear to be not properly compensated for the series resistance. and some action potentials (MPs) are of unusually small amplitude. Please provide the series and/or access resistance for all recordings, and the cut-off for analysis.

"Access resistance was monitored throughout the entire recording. Recordings with access resistance either exceeding 35 MΩ or changed >30% were discarded from further analysis." (Methods, page 6)

The averaged access resistance was 18.32±7.32 (mean ± SD).

AP amplitude shortens because of high frequency firing or because drug applications. For example, paxilline can change AP half-width, AHP and shorten APs (Brenner *et al.*, 2005; Shruti *et al.*, 2008).

Here we provide access resistance for all cells, in all experiments:

15.3	17.4	7.4	12.3	6	19.7	10.5	20	12.2	24.5	22.1	17.3	7	12.9	7	32.3
17.5	21.5	9.6	16.7	8	28.5	8.6	32	30.9	16.1	18.7	15.2	7	16	18.9	14.1
15.1	23.7	9.6	13.8	9.6	11	17.9	30	23.4	25.9	17	10.1	14.4	23.9	21	14.7

20.2	19.7	8.3	22.6	18.6	20.5	15.4	22	28	17.8	23	17.7	14	7.6	14.2	20.1
23.2	21.3	12.2	15	10.4	33.3	23.7	14.8	23.5	27.3	17.3	35	9.9	25.2	8.6	14.4
21.5	10.8	13	18.8	17.8	10.3	7.7	31.4	25.1	14.9	13.2	13.9	16	21	10.3	18.7
15.2	19.8	11.1	31	10.2	15	7.3	16	28.6	24.7	19	15.3	19.2	11.7	18	20.2
32.4	15.7	7.6	19.2	21.6	13.1	5.7	35	20.7	34.2	9	20.7	14.2	15.1	13.2	22.5
34	22	7.5	11.4	24.1	26.9	13.6	35	34.9	10.8	31	25.5	12.1	18.3	29	22
19.2	16.7	15.6	31	29	8.5	26.7	20	13.2	16	11.7	13.4	20.5	14.6	24.8	21.5
20	32.1	18.8	10.6	5.7	8.8	10.4	19.5	26	24	16.2	28	22.3	28.1	20.2	18.1

6. In the figure, many of the comparison histograms show no significant differences. I suggest to relegate at least some of those to the supplemental material.

Unfortunately, supplementary files are not allowed for J. Physiology. For this reason, we removed some data from figures, especially when there were no differences (steepness and midpoint graphs). Description of these parameters remained in the main text with the mean \pm SD and statistical analysis.

7. The comparison of 'regular' ACSF (2.5 mM Ca²⁺) with low Ca²⁺ ACSF is probably exaggerating the effect described. Firstly, 2.5 mM Ca²⁺ is higher than the commonly used ACSF (2 mM Ca²⁺) and too low in the 1 mM Ca²⁺ concentration in the 'Low Ca²⁺' ACSF. In fact, 1 mM Ca²⁺ will result in ~0.8 mM free Ca²⁺. Have the authors looked at direct effects (such as a depolarisation). However, in experiments on human tissue realistic values for the free extracellular Ca²⁺ concentration in the CSF have been measured with ion-sensitive microelectrodes to be between 1.1 to 1.3 mM, i.e 1.3 to 1.5 total Ca²⁺. It would be of interest to now whether the GABA_BR effect is excitatory or inhibitory under this condition. Could the authors please comment!

We performed experiments in more human-like ACSF, with 1.5 mM Ca²⁺. Interestingly, GABA_BR pharmacological activation did not change intrinsic excitability of neither of VIP-IN types (see figure below). However, we do not want to include these data to our manuscript because these results require in-depth analysis. We think that it is an issue that should be dissected in the future experiments. There is no data on how GABA_BRs will modulate activity of different neuronal populations in 1.5 mM Ca²⁺. Since many channels and signaling pathways are Ca²⁺ sensitive, intrinsic excitability and network activity will be shaped by resultant of several factors. Also, modulatory effects of GABA_BRs can be direct or indirect on neuronal excitability.

Figure. L2/3 VIP-IN intrinsic excitability in human-like ACSF conditions (1.5 mM Ca^{2+} ACSF) is not regulated by GABA_ARs. **A)** Example of traces of Type 1 VIP-IN firing responses upon 500 ms-long depolarizing current injection. Lower traces show step intensities of the current injected to the cell, middle traces represent neuronal firings at the rheobase level, upper traces show maximum spike frequency before (Ctrl, *left*) and after the GABA_AR agonist (baclofen, Bac, *right*). **B)** Mean (\pm standard deviation, SD) firing frequency – current injection (F-I) curves for Type 1

VIP-INs in control ACSF and after baclofen ($p > 0.05$, paired t-test, $n = 11$). C) Hill sigmoidal curves fitted to the data showed in B, $n = 11$. D) Within cell comparison and mean (\pm SD) maximum firing frequency, rheobase, steepness and midpoint of Type 1 in control and after baclofen ($p = 0.147$ Wilcoxon test, $p = 0.553$, $p = 0.187$ and $p = 0.047$ paired t-test, respectively, $n = 11$). E-H) Same as A-D but for pooled raw data of Types 2 and 3 VIP-INs ($p = 0.129$, $p = 0.370$, $p = 0.153$ and $p = 0.251$ paired t-test, respectively, $n = 11$). * $p < 0.05$.

8. Page 7, 'Curve fitting' Which data were used for curve fitting?

For curve fitting we used raw data provided for F-I curves. We incorporated this information to the text:

“For the analysis of intrinsic excitability, a frequency-to-intensity plot (F-I curve) was created for each neuron in control condition and after drug administration. Subsequently, using the same raw data, a sigmoidal curve was fitted to every raw F-I plot, employing sigmoidal hill function with the following equation:” (Methods, page 8)

9. Page 7, last para. Please quote the original papers for cluster analysis, not only a review paper.

As suggested, now we cite original papers: (Breiman, 2001), (Wold *et al.*, 1987), (Jolliffe & Cadima, 2016).

10. Page 8, last para. Please quote the original paper when you introduce 'Ward's method'.

Now we refer to the original Ward's paper: (Ward, 1963)

11. Page 14, 2nd paragraph I good way to identify whether the GABA_BR effects were mediated via pre- and/or postsynaptic receptors. Has this been tried? If yes, this should be mentioned here.

We are not sure about this comment. We analyzed how GABA_BRs influence excitatory synaptic transmission to VIP-INs (Figure 9). Baclofen reduced sEPSC frequency suggesting presynaptic effect. We also tested how these receptors modulate inhibitory synaptic transmission to VIP-INs. Here, also baclofen reduced sIPSC frequency suggesting presynaptic effect. However, in case of sIPSCs we used Cs-based internal solution, so we were unable to perform clustering of VIP-INs. We did not include data regarding sIPSCs in the current manuscript.

12. Page 17, last two paragraphs It appears that the effect of GABA_BR if pre- and or has indeed been analysed but no figure is shown. The relevant figure is in the Supplemental Materials. It should be moved to the main text body,

All figures are in the main text now.

References:

Breiman L (2001). Random Forests. *Machine Learning* **45**, 5–32.

- Brenner R, Chen QH, Vilaythong A, Toney GM, Noebels JL & Aldrich RW (2005). BK channel $\beta 4$ subunit reduces dentate gyrus excitability and protects against temporal lobe seizures. *Nat Neurosci* **8**, 1752–1759.
- Dudai A, Yayon N, Lerner V, Tasaka G, Deitcher Y, Gorfine K, Niederhoffer N, Mizrahi A, Soreq H & London M (2020). Barrel cortex VIP/ChAT interneurons suppress sensory responses in vivo ed. Petersen CCH. *PLoS Biol* **18**, e3000613.
- Garaycochea J & Slaughter MM (2016). GABA_B receptors enhance excitatory responses in isolated rat retinal ganglion cells: GABA_B Rs enhance excitatory responses. *J Physiol* **594**, 5543–5554.
- Hioki H, Sohn J, Nakamura H, Okamoto S, Hwang J, Ishida Y, Takahashi M & Kameda H (2018). Preferential inputs from cholecystokinin-positive neurons to the somatic compartment of parvalbumin-expressing neurons in the mouse primary somatosensory cortex. *Brain Research* **1695**, 18–30.
- Jiang S-N, Cao J-W, Liu L-Y, Zhou Y, Shan G-Y, Fu Y-H, Shao Y-C & Yu Y-C (2023). *Sncg*, *Mybpc1*, and *Parm1* Classify subpopulations of VIP-expressing interneurons in layers 2/3 of the somatosensory cortex. *Cerebral Cortex* **33**, 4293–4304.
- Jolliffe IT & Cadima J (2016). Principal component analysis: a review and recent developments. *Phil Trans R Soc A* **374**, 20150202.
- Kawaguchi Y (1997). GABAergic cell subtypes and their synaptic connections in rat frontal cortex. *Cerebral Cortex* **7**, 476–486.
- Maffei A, Nelson SB & Turrigiano GG (2004). Selective reconfiguration of layer 4 visual cortical circuitry by visual deprivation. *Nat Neurosci* **7**, 1353–1359.
- Muñoz-Manchado AB, Bengtsson Gonzales C, Zeisel A, Munguba H, Bekkouche B, Skene NG, Lönnerberg P, Ryge J, Harris KD, Linnarsson S & Hjerling-Leffler J (2018). Diversity of Interneurons in the Dorsal Striatum Revealed by Single-Cell RNA Sequencing and PatchSeq. *Cell Reports* **24**, 2179-2190.e7.
- Nassar M, Richevaux L, Lim D, Tayupo D, Martin E & Fricker D (2024). Presubicular VIP expressing interneurons receive facilitating excitation from anterior thalamus. *Neuroscience* S0306452224004846.
- Ramakrishna Y & Sadeghi SG (2020). Activation of GABA_B receptors results in excitatory modulation of calyx terminals in rat semicircular canal cristae. *Journal of Neurophysiology* **124**, 962–972.
- Sanchez-Vives MV & McCormick DA (2000). Cellular and network mechanisms of rhythmic recurrent activity in neocortex. *Nat Neurosci* **3**, 1027–1034.
- Shruti S, Clem RL & Barth AL (2008). A seizure-induced gain-of-function in BK channels is associated with elevated firing activity in neocortical pyramidal neurons. *Neurobiology of Disease* **30**, 323–330.
- Ward JH (1963). Hierarchical Grouping to Optimize an Objective Function. *Journal of the American Statistical Association* **58**, 236–244.
- Wold S, Esbensen K & Geladi P (1987). Principal component analysis. *Chemometrics and Intelligent Laboratory Systems* **2**, 37–52.
- Xu R & Wunsch II D (2005). Survey of Clustering Algorithms. *IEEE Trans Neural Netw* **16**, 645–678.

Dear Dr Urban-Ciecko,

Re: JP-RP-2024-286439R1 "Inhibition of BK channels by GABAB receptors enhances intrinsic excitability of layer 2/3 VIP-interneurons in mouse neocortex" by Karolina Bogaj and Joanna Urban-Ciecko

We are pleased to tell you that your paper has been accepted for publication in The Journal of Physiology.

Yours sincerely,

Katalin Toth
Senior Editor
The Journal of Physiology

If you would like to receive our 'Research Roundup', a monthly newsletter highlighting the cutting-edge research published in The Physiological Society's family of journals (The Journal of Physiology, Experimental Physiology, Physiological Reports, The Journal of Nutritional Physiology and The Journal of Precision Medicine: Health and Disease), please click this link, fill in your name and email address and select 'Research Roundup':
<https://www.physoc.org/journals-and-media/membernews>

- You can help your research get the attention it deserves! Check out Wiley's free Promotion Guide for best-practice recommendations for promoting your work at: www.wileyauthors.com/eo/guide. You can learn more about Wiley Editing Services which offers professional video, design, and writing services to create shareable video abstracts, infographics, conference posters, lay summaries, and research news stories for your research at: www.wileyauthors.com/eo/promotion.

Reviewing Editor's comments:

The authors have substantially changed their manuscript in response to the reviewer's initial assessment. Both reviewers are now entirely satisfied with the changes and with the overall manuscript.

Referee #1:

Thanks you for your resubmission and addressing my initial concerns. The quality and quantity of your data analysis and the elegance of your experimental design is to be envied.

One minor change. In Figure 10b, you should label the panel is in the presence of synaptic block with AMPA and NMDA antagonists DNQX and D-APV. You do this throughout the remaining parts of Fig10, just not Figure 10b.

Referee #2:

The authors have satisfactorily addressed all my concerns

END OF COMMENTS